# A Probabilistic Framework for Deep Learning

**Ankit B. Patel**
Baylor College of Medicine, Rice University
ankitp@bcm.edu,abp4@rice.edu

**Tan Nguyen**
Rice University
mn15@rice.edu

**Richard G. Baraniuk**
Rice University
richb@rice.edu

## Abstract

We develop a probabilistic framework for deep learning based on the *Deep Rendering Mixture Model* (DRMM), a new *generative probabilistic model* that explicitly capture variations in data due to latent task nuisance variables. We demonstrate that max-sum inference in the DRMM yields an algorithm that exactly reproduces the operations in deep convolutional neural networks (DCNs), providing a first principles derivation. Our framework provides new insights into the successes and shortcomings of DCNs as well as a principled route to their improvement. DRMM training via the Expectation-Maximization (EM) algorithm is a powerful alternative to DCN back-propagation, and initial training results are promising. Classification based on the DRMM and other variants outperforms DCNs in supervised digit classification, training 2-3$\times$ faster while achieving similar accuracy. Moreover, the DRMM is applicable to semi-supervised and unsupervised learning tasks, achieving results that are state-of-the-art in several categories on the MNIST benchmark and comparable to state of the art on the CIFAR10 benchmark.

## 1 Introduction

Humans are adept at a wide array of complicated sensory inference tasks, from recognizing objects in an image to understanding phonemes in a speech signal, despite significant variations such as the position, orientation, and scale of objects and the pronunciation, pitch, and volume of speech. Indeed, the main challenge in many sensory perception tasks in vision, speech, and natural language processing is a high amount of such *nuisance variation*. Nuisance variations complicate perception by turning otherwise simple statistical inference problems with a small number of variables (e.g., class label) into much higher-dimensional problems. The key challenge in developing an inference algorithm is then *how to factor out all of the nuisance variation in the input*. Over the past few decades, a vast literature that approaches this problem from myriad different perspectives has developed, but the most difficult inference problems have remained out of reach.

Recently, a new breed of machine learning algorithms have emerged for high-nuisance inference tasks, achieving super-human performance in many cases. A prime example of such an architecture is the *deep convolutional neural network* (DCN), which has seen great success in tasks like visual object recognition and localization, speech recognition and part-of-speech recognition.

The success of deep learning systems is impressive, but a fundamental question remains: *Why do they work?* Intuitions abound to explain their success. Some explanations focus on properties of feature invariance and selectivity developed over multiple layers, while others credit raw computational power and the amount of available training data. However, beyond these intuitions, a coherent theoretical framework for understanding, analyzing, and synthesizing deep learning architectures has remained elusive.

In this paper, we develop a new theoretical framework that provides insights into both the successes and shortcomings of deep learning systems, as well as a principled route to their design and improvement. Our framework is based on a *generative probabilistic model that explicitly captures variation due to latent nuisance variables*. The *Rendering Mixture Model* (RMM) explicitly models nuisance variation through a *rendering function* that combines task target variables (e.g., object class in an

object recognition) with a collection of task nuisance variables (e.g., pose). The *Deep Rendering Mixture Model* (DRMM) extends the RMM in a hierarchical fashion by rendering via a product of affine nuisance transformations across multiple levels of abstraction. The graphical structures of the RMM and DRMM enable efficient inference via message passing (e.g., using the max-sum/product algorithm) and training via the expectation-maximization (EM) algorithm. A key element of our framework is the relaxation of the RMM/DRMM generative model to a discriminative one in order to optimize the bias-variance tradeoff. Below, we demonstrate that the computations involved in joint MAP inference in the relaxed DRMM coincide exactly with those in a DCN.

The intimate connection between the DRMM and DCNs provides a range of new insights into how and why they work and do not work. While our theory and methods apply to a wide range of different inference tasks (including, for example, classification, estimation, regression, etc.) that feature a number of task-irrelevant nuisance variables (including, for example, object and speech recognition), for concreteness of exposition, we will focus below on the classification problem underlying visual object recognition. The proofs of several results appear in the Appendix.

## 2 Related Work

**Theories of Deep Learning.** Our theoretical work shares similar goals with several others such as the *i*-Theory [1] (one of the early inspirations for this work), Nuisance Management [24], the Scattering Transform [6], and the simple sparse network proposed by Arora et al. [2].

**Hierarchical Generative Models.** The DRMM is closely related to several hierarchical models, including the Deep Mixture of Factor Analyzers [27] and the Deep Gaussian Mixture Model [29].

Like the above models, the DRMM attempts to employ parameter sharing, capture the notion of nuisance transformations explicitly, learn selectivity/invariance, and promote sparsity. However, the key features that distinguish the DRMM approach from others are: (i) The DRMM explicitly models nuisance variation across multiple levels of abstraction via a product of affine transformations. This factorized linear structure serves dual purposes: it enables (ii) tractable inference (via the max-sum/product algorithm), and (iii) it serves as a regularizer to prevent overfitting by an exponential reduction in the number of parameters. Critically, (iv) inference is not performed for a single variable of interest but instead for the full global configuration of nuisance variables. This is justified in low-noise settings. And most importantly, (v) we can derive the structure of DCNs *precisely*, endowing DCN operations such as the convolution, rectified linear unit, and spatial max-pooling with principled probabilistic interpretations. Independently from our work, Soatto et al. [24] also focus strongly on nuisance management as the key challenge in defining good scene representations. However, their work considers max-pooling and ReLU as *approximations* to a marginalized likelihood, whereas our work interprets those operations differently, in terms of max-sum inference in a specific probabilistic generative model. The work on the number of linear regions in DCNs [14] is complementary to our own, in that it sheds light on the complexity of functions that a DCN can compute. Both approaches could be combined to answer questions such as: How many templates are required for accurate discrimination? How many samples are needed for learning? We plan to pursue these questions in future work.

**Semi-Supervised Neural Networks.** Recent work in neural networks designed for semi-supervised learning (few labeled data, lots of unlabeled data) has seen the resurgence of generative-like approaches, such as Ladder Networks [17], Stacked What-Where Autoencoders (SWWAE) [31] and many others. These network architectures augment the usual task loss with one or more regularization term, typically including an image reconstruction error, and train jointly. A key difference with our DRMM-based approach is that these networks do not arise from a proper probabilistic density and as such they must resort to learning the bottom-up recognition and top-down reconstruction weights separately, and they cannot keep track of uncertainty.

## 3 The Deep Rendering Mixture Model: Capturing Nuisance Variation

Although we focus on the DRMM in this paper, we define and explore several other interesting variants, including the Deep Rendering Factor Model (DRFM) and the Evolutionary DRMM (E-DRMM), both of which are discussed in more detail in [16] and the Appendix. The E-DRMM is particularly important, since its max-sum inference algorithm yields a decision tree of the type employed in a random decision forest classifier[5].

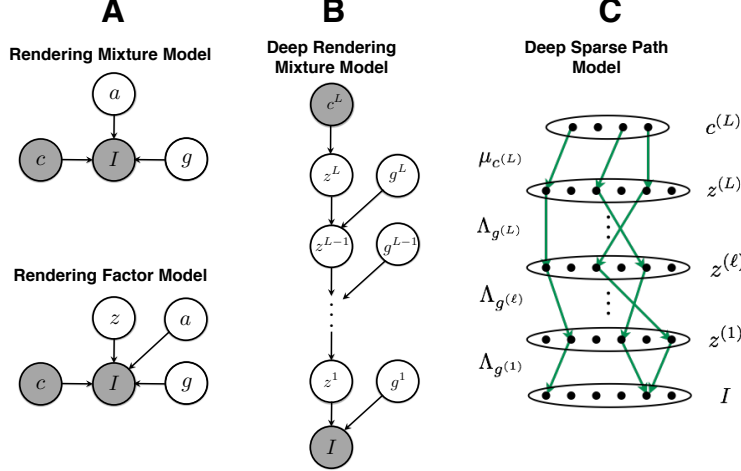

Figure 1: Graphical model depiction of (A) the Shallow Rendering Models and (B) the DRMM. All dependence on pixel location $x$ has been suppressed for clarity. (C) The Sparse Sum-over-Paths formulation of the DRMM. A rendering path contributes only if it is active (green arrows).

### 3.1 The (Shallow) Rendering Mixture Model

The RMM is a *generative probabilistic model* for images that explicitly models the relationship between images $I$ of the same object $c$ subject to nuisance $g \in \mathcal{G}$, where $\mathcal{G}$ is the set of all nuisances (see Fig. 1A for the graphical model depiction).

$$c \sim \mathrm{Cat}(\{\pi_c\}_{c \in \mathcal{C}}), \quad g \sim \mathrm{Cat}(\{\pi_g\}_{g \in \mathcal{G}}), \quad a \sim \mathrm{Bern}(\{\pi_a\}_{a \in \mathcal{A}}),$$
$$I = a\mu_{cg} + \text{ noise}. \tag{1}$$

Here, $\mu_{cg}$ is a template that is a function of the class $c$ and the nuisance $g$. The switching variable $a \in \mathcal{A} = \{\text{ON, OFF}\}$ determines whether or not to render the template at a particular patch; a sparsity prior on $a$ thus encourages each patch to have a few causes. The noise distribution is from the exponential family, but without loss of generality we illustrate below using Gaussian noise $\mathcal{N}(0, \sigma^2 \mathbf{1})$. We assume that the noise is i.i.d. as a function of pixel location $x$ and that the class and nuisance variables are independently distributed according to categorical distributions. (Independence is merely a convenience for the development; in practice, $g$ can depend on $c$.) Finally, since the world is spatially varying and an image can contain a number of different objects, it is natural to break the image up into a number of *patches*, that are centered on a single pixel $x$. The RMM described in (1) then applies at the patch level, where $c$, $g$, and $a$ depend on pixel/patch location $x$. We will omit the dependence on $x$ when it is clear from context.

**Inference in the Shallow RMM Yields One Layer of a DCN.** We now connect the RMM with the computations in one layer of a deep convolutional network (DCN). To perform object recognition with the RMM, we must marginalize out the nuisance variables $g$ and $a$. Maximizing the log-posterior over $g \in \mathcal{G}$ and $a \in \mathcal{A}$ and then choosing the most likely class yields the *max-sum classifier*

$$\hat{c}(I) = \underset{c \in \mathcal{C}}{\mathrm{argmax}} \max_{g \in \mathcal{G}} \max_{a \in \mathcal{A}} \ln p(I|c, g, a) + \ln p(c, g, a) \tag{2}$$

that computes the most likely global configuration of target and nuisance variables for the image. Assuming that Gaussian noise is added to the template, the image is normalized so that $\|I\|^2 = 1$, and $c, g$ are uniformly distributed, (2) becomes

$$\hat{c}(I) \equiv \underset{c \in \mathcal{C}}{\mathrm{argmax}} \max_{g \in \mathcal{G}} \max_{a \in \mathcal{A}} a(\langle w_{cg}|I \rangle + b_{cg}) + b_a = \underset{c \in \mathcal{C}}{\mathrm{argmax}} \max_{g \in \mathcal{G}} \mathrm{ReLu}(\langle w_{cg}|I \rangle + b_{cg}) + b_0 \tag{3}$$

where $\mathrm{ReLU}(u) \equiv (u)_+ = \max\{u, 0\}$ is the soft-thresholding operation performed by the rectified linear units in modern DCNs. Here we have reparameterized the RMM model from the *moment parameters* $\theta \equiv \{\sigma^2, \mu_{cg}, \pi_a\}$ to the *natural parameters* $\eta(\theta) \equiv \{w_{cg} \equiv \frac{1}{\sigma^2}\mu_{cg}, b_{cg} \equiv -\frac{1}{2\sigma^2}\|\mu_{cg}\|_2^2, b_a \equiv \ln p(a) = \ln \pi_a, b_0 \equiv \ln\left(\frac{p(a=1)}{p(a=0)}\right)$. The relationships $\eta(\theta)$ are referred to as the *generative parameter constraints*.

We now demonstrate that *the sequence of operations in the max-sum classifier in (3) coincides exactly with the operations involved in one layer of a DCN*: image normalization, linear template matching, thresholding, and max pooling. First, the image is *normalized* (by assumption). Second, the image is filtered with a set of noise-scaled rendered templates $w_{cg}$. If we assume *translational invariance* in the RMM, then the rendered templates $w_{cg}$ yield a *convolutional* layer in a DCN [10] (see Appendix Lemma A.2). Third, the resulting activations (log-probabilities of the hypotheses) are passed through a pooling layer; if $g$ is a translational nuisance, then taking the maximum over $g$ corresponds to *max pooling* in a DCN. Fourth, since the switching variables are latent (unobserved), we max-marginalize over them during classification. This leads to the ReLU operation (see Appendix Proposition A.3).

## 3.2 The Deep Rendering Mixture Model: Capturing Levels of Abstraction

Marginalizing over the nuisance $g \in \mathcal{G}$ in the RMM is intractable for modern datasets, since $\mathcal{G}$ will contain all configurations of the high-dimensional nuisance variables $g$. In response, we extend the RMM into a hierarchical *Deep Rendering Mixture Model* (DRMM) by factorizing $g$ into a number of different nuisance variables $g^{(1)}, g^{(2)}, \ldots, g^{(L)}$ at different levels of abstraction. The DRMM image generation process starts at the highest level of abstraction ($\ell = L$), with the random choice of the object class $c^{(L)}$ and overall nuisance $g^{(L)}$. It is then followed by random choices of the lower-level details $g^{(\ell)}$ (we absorb the switching variable $a$ into $g$ for brevity), progressively rendering more concrete information level-by-level ($\ell \rightarrow \ell-1$), until the process finally culminates in a fully rendered $D$-dimensional image $I$ ($\ell = 0$). Generation in the DRMM takes the form:

$$c^{(L)} \sim \text{Cat}(\{\pi_{c^{(L)}}\}), \ g^{(\ell)} \sim \text{Cat}(\{\pi_{g^{(\ell)}}\}) \ \forall \ell \in [L] \tag{4}$$

$$\mu_{c^{(L)}g} \equiv \Lambda_g \mu_{c^{(L)}} \equiv \Lambda^{(1)}_{g^{(1)}} \Lambda^{(2)}_{g^{(2)}} \cdots \Lambda^{(L-1)}_{g^{(L-1)}} \Lambda^{(L)}_{g^{(L)}} \mu_{c^{(L)}} \tag{5}$$

$$I \sim \mathcal{N}(\mu_{c^{(L)}g}, \Psi \equiv \sigma^2 \mathbf{1}_D), \tag{6}$$

where the latent variables, parameters, and helper variables are defined in full detail in Appendix B.

The DRMM is a deep Gaussian Mixture Model (GMM) with special constraints on the latent variables. Here, $c^{(L)} \in \mathcal{C}^L$ and $g^{(\ell)} \in \mathcal{G}^\ell$, where $\mathcal{C}^L$ is the set of target-relevant nuisance variables, and $\mathcal{G}^\ell$ is the set of all target-irrelevant nuisance variables at level $\ell$. The *rendering path* is defined as the sequence $(c^{(L)}, g^{(L)}, \ldots, g^{(\ell)}, \ldots, g^{(1)})$ from the root (overall class) down to the individual pixels at $\ell = 0$. $\mu_{c^{(L)}g}$ is the template used to render the image, and $\Lambda_g \equiv \prod_\ell \Lambda_{g^{(\ell)}}$ represents the sequence of local nuisance transformations that partially render finer-scale details as we move from abstract to concrete. Note that each $\Lambda^{(\ell)}_{g^{(\ell)}}$ is an *affine* transformation with a bias term $\alpha^{(\ell)}_{g^{(\ell)}}$ that we have suppressed for clarity. Fig. 1B illustrates the corresponding graphical model. As before, we have suppressed the dependence of $g^{(\ell)}$ on the pixel location $x^{(\ell)}$ at level $\ell$ of the hierarchy.

**Sum-Over-Paths Formulation of the DRMM.** We can rewrite the DRMM generation process by expanding out the matrix multiplications into scalar products. This yields an interesting new perspective on the DRMM, as each pixel intensity $I_x = \sum_p \lambda^{(L)}_p a^{(L)}_p \cdots \lambda^{(1)}_p a^{(1)}_p$ is the sum over all *active paths* to that pixel, of the product of weights along that path. A rendering path $p$ is active iff every switch on the path is active i.e. $\prod_\ell a^{(\ell)}_p = 1$ . While exponentially many possible rendering paths exist, only a very small fraction, controlled by the sparsity of $a$, are active. Fig. 1C depicts the sum-over-paths formulation graphically.

**Recursive and Nonnegative Forms.** We can rewrite the DRMM into a recursive form as $z^{(\ell)} = \Lambda^{(\ell+1)}_{g^{(\ell+1)}} z^{(\ell+1)}$, where $z^{(L)} \equiv \mu_{c^{(L)}}$ and $z^{(0)} \equiv I$. We refer to the helper latent variables $z^{(\ell)}$ as *intermediate rendered templates*. We also define the *Nonnegative DRMM* (NN-DRMM) as a DRMM with an extra nonnegativity constraint on the intermediate rendered templates, $z^{(\ell)} \geq 0 \forall \ell \in [L]$. The latter is enforced in training via the use of a ReLu operation in the top-down reconstruction phase of inference. Throughout the rest of the paper, we will focus on the NN-DRMM, leaving the unconstrained DRMM for future work. For brevity, we will drop the NN prefix.

**Factor Model.** We also define and explore a variant of the DRMM that where the top-level latent variable is Gaussian: $z^{(L+1)} \sim \mathcal{N}(0, \mathbf{1}_d) \in \mathbb{R}^d$ and the recursive generation process is otherwise identical to the DRMM: $z^{(\ell)} = \Lambda^{(\ell+1)}_{g^{(\ell+1)}} z^{(\ell+1)}$ where $g^{(L+1)} \equiv c^{(L)}$. We call this the *Deep Rendering Factor Model* (DRFM). The DRFM is closely related to the Spike-and-Slab Sparse Coding model [22]. Below we explore some training results, but we leave most of the exploration for future work. (see Fig. 3 in Appendix C for architecture of the RFM, the shallow version of the DRFM)

**Number of Free Parameters.** Compared to the shallow RMM, which has $D\,|\mathcal{C}^L|\prod_\ell|\mathcal{G}^\ell|$ parameters, the DRMM has only $\sum_\ell|\mathcal{G}^{\ell+1}|D^\ell D^{\ell+1}$ parameters, an *exponential reduction in the number of free parameters* (Here $\mathcal{G}^{L+1}\equiv\mathcal{C}^L$ and $D^\ell$ is the number of units in the $\ell$-th layer with $D^0\equiv D$). This enables efficient inference, learning, and better generalization. Note that we have assumed dense (fully connected) $\Lambda_g$'s here; if we impose more structure (e.g. translation invariance), the number of parameters will be further reduced.

**Bottom-Up Inference.** As in the shallow RMM, given an input image $I$ the DRMM classifier infers the most likely global configuration $\{c^{(L)},g^{(\ell)}\}$, $\ell=0,1,\ldots,L$ by executing the max-sum/product message passing algorithm in two stages: (i) bottom-up (from fine-to-coarse) to infer the overall class label $\hat{c}^{(L)}$ and (ii) top-down (from coarse-to-fine) to infer the latent variables $\hat{g}^{(\ell)}$ at all intermediate levels $\ell$. First, we will focus on the fine-to-coarse pass since it leads directly to DCNs.

Using (3), the *fine-to-coarse* NN-DRMM inference algorithm for inferring the most likely cateogry $\hat{c}^L$ is given by

$$\operatorname*{argmax}_{c^{(L)}\in\mathcal{C}}\max_{g\in\mathcal{G}}\mu^T_{c^{(L)}}g\,I=\operatorname*{argmax}_{c^{(L)}\in\mathcal{C}}\max_{g\in\mathcal{G}}\mu^T_{c^{(L)}}\prod_{\ell=L}^{1}\Lambda^T_{g^{(\ell)}}I$$

$$=\operatorname*{argmax}_{c^{(L)}\in\mathcal{C}}\mu^T_{c^{(L)}}\max_{g^{(L)}\in\mathcal{G}^L}\Lambda^T_{g^{(L)}}\cdots\underbrace{\max_{g^{(1)}\in\mathcal{G}^1}\Lambda^T_{g^{(1)}}|I}_{\equiv I^1}\;=\;\cdots\;\equiv\operatorname*{argmax}_{c^{(L)}\in\mathcal{C}}\mu^T_{c^{(L)}}I^{(L)}.\quad(7)$$

Here, we have assumed the bias terms $\alpha_{g^{(\ell)}}=0$. In the second line, we used the max-product algorithm (distributivity of max over products i.e. for $a>0$, $\max\{ab,ac\}=a\max\{b,c\}$). See Appendix B for full details. This enables us to rewrite (7) recursively:

$$I^{(\ell+1)}\equiv\max_{g^{(\ell+1)}\in\mathcal{G}^{\ell+1}}\underbrace{(\Lambda_{g^{(\ell+1)}})^T}_{\equiv W^{(\ell+1)}}I^{(\ell)}=\text{MaxPool}(\text{ReLu}(\text{Conv}(I^{(\ell)}))),\quad(8)$$

where $I^{(\ell)}$ is the output *feature maps* of layer $\ell$, $I^{(0)}\equiv I$ and $W^{(\ell)}$ are the filters/weights for layer $\ell$. Comparing to (3), we see that the $\ell$-th iteration of (7) and (8) corresponds to feedforward propagation in the $\ell$-th layer of a DCN. *Thus a DCN's operation has a probabilistic interpretation as fine-to-coarse inference of the most probable configuration in the DRMM.*

**Top-Down Inference.** A unique contribution of our generative model-based approach is that we have a principled derivation of a top-down inference algorithm for the NN-DRMM (Appendix B). The resulting algorithm amounts to a simple top-down reconstruction term $\hat{I}_n=\Lambda_{\hat{g}_n}\mu_{\hat{c}_n^{(L)}}$.

**Discriminative Relaxations: From Generative to Discriminative Classifiers.** We have constructed a correspondence between the DRMM and DCNs, but the mapping is not yet complete. In particular, recall the generative constraints on the weights and biases. DCNs do not have such constraints — their weights and biases are free parameters. As a result, when faced with training data that violates the DRMM's underlying assumptions, the DCN will have more freedom to compensate. In order to complete our mapping from the DRMM to DCNs, we *relax* these parameter constraints, allowing the weights and biases to be free and independent parameters. We refer to this process as a *discriminative relaxation of a generative classifier* ([15, 4], see the Appendix D for details).

### 3.3 Learning the Deep Rendering Model via the Expectation-Maximization (EM) Algorithm

We describe how to learn the DRMM parameters from training data via the hard EM algorithm in Algorithm 1. The DRMM E-Step consists of bottom-up and top-down (reconstruction) E-steps at each layer $\ell$ in the model. The $\gamma_{ncg}\equiv p(c,g|I_n;\theta)$ are the responsibilities, where for brevity we have absorbed $a$ into $g$. The DRMM M-step consists of M-steps for each layer $\ell$ in the model. The per-layer M-step in turn consists of a responsibility-weighted regression, where $\text{GLS}(y_n\sim x_n)$ denotes the solution to a generalized Least Squares regression problem that predict targets $y_n$ from predictors $x_n$ and is closely related to the SVD. The Iversen bracket is defined as $[\![b]\!]\equiv 1$ if expression $b$ is true and is 0 otherwise. There are several interesting and useful features of the EM algorithm. First, we note that it is a *derivative-free alternative to the back propagation algorithm* for training that is both intuitive and potentially much faster (provided a good implementation for the GLS problem). Second, it is easily parallelized over layers, since the M-step updates each layer separately (model parallelism). Moreover, it can be extended to a batch version so that at each iteration the model is

---

**Algorithm 1** Hard EM and EG Algorithms for the DRMM

$$\textbf{E-step:} \qquad \hat{c}_n, \hat{g}_n = \underset{c,g}{\operatorname{argmax}} \; \gamma_{ncg}$$

$$\textbf{M-step:} \qquad \hat{\Lambda}_{g^{(\ell)}} = \underbrace{\text{GLS}}_{} \left( I_n^{(\ell-1)} \sim \hat{z}_n^{(\ell)} \mid g^{(\ell)} = \hat{g}_n^{(\ell)} \right) \; \forall g^{(\ell)}$$

$$\textbf{G-step:} \qquad \Delta \hat{\Lambda}_{g^{(\ell)}} \propto \nabla_{\Lambda_{g^{(\ell)}}} \ell_{DRMM}(\theta)$$

---

simultaneously updated using separate subsets of the data (data parallelism). This will enable training to be distributed easily across multiple machines. In this vein, our EM algorithm shares several features with the ADMM-based Bregman iteration algorithm in [28]. However, the motivation there is from an optimization perspective and so the resulting training algorithm is not derived from a proper probabilistic density. Third, it is far more interpretable via its connections to (deep) sparse coding and to the hard EM algorithm for GMMs. The sum-over-paths formulation makes it particularly clear that the mixture components are paths (from root to pixels) in the DRMM.

**G-step.** For the training results in this paper, we use the Generalized EM algorithm wherein we replace the M-step with a gradient descent based G-step (see Algorithm 1). This is useful for comparison with backpropagation-based training and for ease of implementation.

**Flexibility and Extensibility.** Since we can choose different priors/types for the nuisances $g$, the larger DRMM family could be useful for modeling a wider range of inputs, including scenes, speech and text. The EM algorithm can then be used to train the whole system end-to-end on different sources/modalities of labeled and unlabeled data. Moreover, the capability to sample from the model allows us to probe what is captured by the DRMM, providing us with principled ways to improve the model. And finally, in order to properly account for noise/uncertainty, it is possible in principle to extend this algorithm into a *soft* EM algorithm. We leave these interesting extensions for future work.

### 3.4 New Insights into Deep Convnets

**DCNs are Message Passing Networks.** The *convolution, Max-Pooling and ReLu operations in a DCN correspond to max-sum/product inference in a DRMM*. Note that by "max-sum-product" we mean a novel combination of max-sum and max-product as described in more detail in the proofs in the Appendix. Thus, we see that architectures and layer types commonly used in today's DCNs can be derived from precise probabilistic assumptions that entirely determine their structure. The DRMM therefore unifies two perspectives — neural network and probabilistic inference (see Table 2 in the Appendix for details).

**Shortcomings of DCNs.** DCNs perform poorly in categorizing transparent objects [20]. This might be explained by the fact that transparent objects generate pixels that have multiple sources, conflicting with the DRMM sparsity prior on $a$, which encourages few sources. DCNs also fail to classify slender and man-made objects [20]. This is because of the locality imposed by the locally-connected/convolutional layers, or equivalently, the small size of the template $\mu_{c^{(L)}g}$ in the DRMM. As a result, DCNs fail to model long-range correlations.

**Class Appearance Models and Activity Maximization.** The DRMM enables us to understand how trained DCNs distill and store knowledge from past experiences in their parameters. Specifically, the DRMM generates rendered templates $\mu_{c^{(L)}g}$ via a mixture of products of affine transformations, thus implying that *class appearance models in DCNs are stored in a similar factorized-mixture form over multiple levels of abstraction.* As a result, it is the product of all the filters/weights over all layers that yield meaningful images of objects (Eq. 6). We can also shed new light on another approach to understanding DCN memories that proceeds by searching for input images that maximize the activity of a particular class unit (say, class of cats) [23], a technique we call *activity maximization*. Results from activity maximization on a high performance DCN trained on 15 million images is shown in Fig. 1 of [23]. The resulting images reveal much about how DCNs store memories. Using the DRMM, the solution $I_{c^{(L)}}^*$ of the activity maximization for class $c^{(L)}$ can be derived as the sum of individual activity-maximizing patches $I_{\mathcal{P}_i}^*$, each of which is a function of the learned DRMM parameters (see Appendix E). In particular, $I_{c^{(L)}}^* \equiv \sum_{\mathcal{P}_i \in \mathcal{P}} I_{\mathcal{P}_i}^*(c^{(L)}, g_{\mathcal{P}_i}^*) \propto \sum_{\mathcal{P}_i \in \mathcal{P}} \mu(c^{(L)}, g_{\mathcal{P}_i}^*)$.

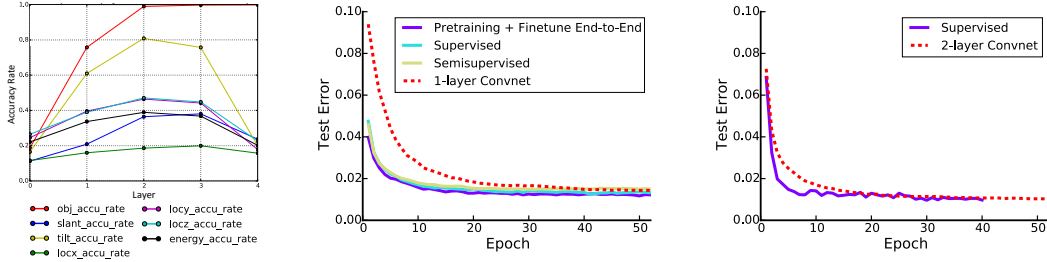

Figure 2: Information about latent nuisance variables at each layer (Left), training results from EG for RFM (Middle) and DRFM (Right) on MNIST, as compared to DCNs of the same configuration.

This implies that $T^*_{c^{(L)}}$ contains multiple appearances of the same object but in various poses. Each activity-maximizing patch has its own pose $g^*_{\mathcal{P}_i}$, consistent with Fig. 1 of [23] and our own extensive experiments with AlexNet, VGGNet, and GoogLeNet (data not shown). Such images provide strong confirmational evidence that the underlying model is a mixture over nuisance parameters, as predcted by the DRMM.

**Unsupervised Learning of Latent Task Nuisances.** A key goal of representation learning is to disentangle the factors of variation that contribute to an image's appearance. Given our formulation of the DRMM, it is clear that DCNs are discriminative classifiers that capture these factors of variation with latent nuisance variables $g$. As such, the theory presented here makes a clear prediction that *for a DCN, supervised learning of task targets will lead to unsupervised learning of latent task nuisance variables*. From the perspective of manifold learning, this means that the architecture of DCNs is designed to learn and disentangle the intrinsic dimensions of the data manifolds.

In order to test this prediction, we trained a DCN to classify synthetically rendered images of naturalistic objects, such as cars and cats, with variation in factors such as location, pose, and lighting. After training, we probed the layers of the trained DCN to quantify how much linearly decodable information exists about the task target $c^{(L)}$ and latent nuisance variables $g$. Fig. 2 (Left) shows that the trained DCN possesses significant information about latent factors of variation and, furthermore, the more nuisance variables, the more layers are required to disentangle the factors. This is strong evidence that depth is necessary and that the amount of depth required increases with the complexity of the class models and the nuisance variations.

## 4   Experimental Results

We evaluate the DRMM and DRFM's performance on the MNIST dataset, a standard digit classification benchmark with a training set of 60,000 $28 \times 28$ labeled images and a test set of 10,000 labeled images. We also evaluate the DRMM's performance on CIFAR10, a dataset of natural objects which include a training set of 50,000 $32 \times 32$ labeled images and a test set of 10,000 labeled images. In all experiments, we use a full E-step that has a bottom-up phase and a principled top-down reconstruction phase. In order to approximate the class posterior in the DRMM, we include a Kullback-Leibler divergence term between the inferred posterior $p(c|I)$ and the true prior $p(c)$ as a regularizer [9]. We also replace the M-step in the EM algorithm of Algorithm 1 by a G-step where we update the model parameters via gradient descent. This variant of EM is known as the Generalized EM algorithm [3], and here we refer to it as EG. All DRMM experiments were done with the NN-DRMM. Configurations of our models and the corresponding DCNs are provided in the Appendix I.

**Supervised Training.** Supervised training results are shown in Table 3 in the Appendix. *Shallow RFM:* The 1-layer RFM (RFM sup) yields similar performance to a Convnet of the same configuration (1.21% vs. 1.30% test error). Also, as predicted by the theory of generative vs discriminative classifiers, EG training converges 2-3x faster than a DCN (18 vs. 40 epochs to reach 1.5% test error, Fig. 2, middle). *Deep RFM:* Training results from an initial implementation of the 2-layer DRFM EG algorithm converges $2 - 3\times$ faster than a DCN of the same configuration, while achieving a *similar* asymptotic test error (Fig. 2, Right). Also, for completeness, we compare supervised training for a 5-layer DRMM with a corresponding DCN, and they show comparable accuracy (0.89% vs 0.81%, Table 3).

**Unsupervised Training.** We train the RFM and the 5-layer DRMM unsupervised with $N_U$ images, followed by an end-to-end re-training of the whole model (unsup-pretr) using $N_L$ labeled images. The results and comparison to the SWWAE model are shown in Table 1. The DRMM model outperforms the SWWAE model in both scenarios (Filters and reconstructed images from the RFM are available in the Appendix 4.)

Table 1: Comparison of Test Error rates (%) between best DRMM variants and other best published results on MNIST dataset for the semi-supervised setting (taken from [31]) with $N_U = 60K$ unlabeled images, of which $N_L \in \{100, 600, 1K, 3K\}$ are labeled.

| **Model** | $N_L = 100$ | $N_L = 600$ | $N_L = 1K$ | $N_L = 3K$ |
|---|---|---|---|---|
| Convnet [10] | 22.98 | 7.86 | 6.45 | 3.35 |
| MTC [18] | 12.03 | 5.13 | 3.64 | 2.57 |
| PL-DAE [11] | 10.49 | 5.03 | 3.46 | 2.69 |
| WTA-AE [13] | - | 2.37 | 1.92 | - |
| SWWAE dropout [31] | $8.71 \pm 0.34$ | $3.31 \pm 0.40$ | $2.83 \pm 0.10$ | $2.10 \pm 0.22$ |
| M1+TSVM [8] | $11.82 \pm 0.25$ | 5.72 | 4.24 | 3.49 |
| M1+M2 [8] | $3.33 \pm 0.14$ | $2.59 \pm 0.05$ | $2.40 \pm 0.02$ | $2.18 \pm 0.04$ |
| Skip Deep Generative Model [12] | 1.32 | - | - | - |
| LadderNetwork [17] | $1.06 \pm 0.37$ | - | $\mathbf{0.84 \pm 0.08}$ | - |
| Auxiliary Deep Generative Model [12] | 0.96 | - | - | - |
| catGAN [25] | $1.39 \pm 0.28$ | - | - | - |
| ImprovedGAN [21] | $0.93 \pm 0.065$ | - | - | - |
| RFM | 14.47 | 5.61 | 4.67 | 2.96 |
| DRMM 2-layer semi-sup | 11.81 | 3.73 | 2.88 | 1.72 |
| DRMM 5-layer semi-sup | 3.50 | **1.56** | 1.67 | **0.91** |
| DRMM 5-layer semi-sup NN+KL | **0.57** | — | — | — |
| SWWAE unsup-pretr [31] | - | 9.80 | 6.135 | 4.41 |
| RFM unsup-pretr | 16.2 | 5.65 | 4.64 | 2.95 |
| DRMM 5-layer unsup-pretr | **12.03** | **3.61** | **2.73** | **1.68** |

**Semi-Supervised Training.** For semi-supervised training, we use a randomly chosen subset of $N_L = 100, 600, 1K$, and $3K$ labeled images and $N_U = 60K$ unlabeled images from the training and validation set. Results are shown in Table 1 for a RFM, a 2-layer DRMM and a 5-layer DRMM with comparisons to related work. The DRMMs performs comparably to state-of-the-art models. Specially, the 5-layer DRMM yields the best results when $N_L = 3K$ and $N_L = 600$ while results in the second best result when $N_L = 1K$. We also show the training results of a 9-layer DRMM on CIFAR10 in Table 4 in Appendix H. The DRMM yields comparable results on CIFAR10 with the best semi-supervised methods. For more results and comparison with other works, see Appendix H.

# 5 Conclusions

Understanding successful deep vision architectures is important for improving performance and solving harder tasks. In this paper, we have introduced a new family of hierarchical generative models, whose inference algorithms for two different models reproduce deep convnets and decision trees, respectively. Our initial implementation of the DRMM EG algorithm outperforms DCN back-propagation in both supervised and unsupervised classification tasks and achieves comparable/state-of-the-art performance on several semi-supervised classification tasks, with no architectural hyperparameter tuning.

**Acknowledgments.** Thanks to Xaq Pitkow and Ben Poole for helpful feedback. ABP and RGB were supported by IARPA via DoI/IBC contract D16PC00003. [1] RGB was supported by NSF CCF-1527501, AFOSR FA9550-14-1-0088, ARO W911NF-15-1-0316, and ONR N00014-12-1-0579. TN was supported by an NSF Graduate Reseach Fellowship and NSF IGERT Training Grant (DGE-1250104).

## Footnotes

[1]The U.S. Government is authorized to reproduce and distribute reprints for Governmental purposes notwithstanding any copyright annotation thereon. Disclaimer: The views and conclusions contained herein are those of the authors and should not be interpreted as necessarily representing the official policies or endorsements, either expressed or implied, of IARPA, DoI/IBC, or the U.S. Government.

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
