[Supplementary Material]

## A  From the Rendering Mixture Model Classifier to a DCN Layer

**Proposition A.1** (MaxOut Neural Networks). *The discriminative relaxation of a noise-free Gaussian Rendering Mixture Model (GRMM) classifier with nuisance variable $g \in \mathcal{G}$ is a single layer neural net consisting of a local template matching operation followed by a piecewise linear activation function (also known as a* MaxOut NN *[7]).*

*Proof.* For transparency, we prove this claim exhaustively. Later claims will have simpler proofs. We have

$$
\begin{aligned}
\hat{c}(I) &\equiv \underset{c\in\mathcal{C}}{\operatorname{argmax}}\, p(c|I) \\
&= \underset{c\in\mathcal{C}}{\operatorname{argmax}}\, \{p(I|c)p(c)\} \\
&= \underset{c\in\mathcal{C}}{\operatorname{argmax}}\, \left\{ \sum_{g\in\mathcal{G}} p(I|c,g)p(c,g) \right\} \\
&\overset{(a)}{=} \underset{c\in\mathcal{C}}{\operatorname{argmax}} \left\{ \max_{g\in\mathcal{G}} p(I|c,g)p(c,g) \right\} \\
&= \underset{c\in\mathcal{C}}{\operatorname{argmax}} \left\{ \max_{g\in\mathcal{G}} \exp\left( \ln p(I|c,g) + \ln p(c,g) \right) \right\} \\
&\overset{(b)}{=} \underset{c\in\mathcal{C}}{\operatorname{argmax}} \left\{ \max_{g\in\mathcal{G}} \exp\left( \sum_{\omega} \ln p(I^{\omega}|c,g) + \ln p(c,g) \right) \right\} \\
&\overset{(c)}{=} \underset{c\in\mathcal{C}}{\operatorname{argmax}} \left\{ \max_{g\in\mathcal{G}} \exp\left( -\frac{1}{2}\sum_{\omega} \langle I^{\omega} - \mu_{cg}^{\omega} | \Sigma_{cg}^{-1} | I^{\omega} - \mu_{cg}^{\omega} \rangle + \ln p(c,g) - \frac{D}{2}\ln|\Sigma_{cg}| \right) \right\} \\
&= \underset{c\in\mathcal{C}}{\operatorname{argmax}} \left\{ \max_{g\in\mathcal{G}} \exp\left( \sum_{\omega} \langle w_{cg}^{\omega} | I^{\omega} \rangle + b_{cg}^{\omega} \right) \right\} \\
&\overset{(d)}{\equiv} \underset{c\in\mathcal{C}}{\operatorname{argmax}} \left\{ \exp\left( \max_{g\in\mathcal{G}} \{ w_{cg} \star_{LC} I \} \right) \right\} \\
&= \underset{c\in\mathcal{C}}{\operatorname{argmax}} \left\{ \max_{g\in\mathcal{G}} \{ w_{cg} \star_{LC} I \} \right\} \\
&= \operatorname{Choose}\{\operatorname{MaxOutPool}(\operatorname{LocalTemplateMatch}(I))\} \\
&= \operatorname{MaxOut-NN}(I;\theta).
\end{aligned}
$$

In line (a), we take the noise-free limit of the GRMM, which means that one hypothesis $(c,g)$ dominates all others in likelihood. In line (b), we assume that the image $I$ consists of multiple channels $\omega \in \Omega$, that are conditionally independent given the global configuration $(c,g)$. Typically, for input images these are color channels and $\Omega \equiv \{R, G, B\}$ but in general $\Omega$ can be more abstract (e.g. as in feature maps). In line (c), we assume that the pixel noise covariance is isotropic and conditionally independent given the global configuration $(c,g)$, so that $\Sigma_{cg} = \sigma_x^2 \mathbf{1}_D$ is proportional to the $D \times D$ identity matrix $\mathbf{1}_D$. In line (d), we defined the *locally connected template matching operator* $\star_{LC}$, which is a location-dependent template matching operation. $\qquad\square$

Note that the nuisance variables $g \in \mathcal{G}$ are (max-)marginalized over, after the application of a local template matching operation against a set of filters/templates $\mathcal{W} \equiv \{w_{cg}\}_{c\in\mathcal{C},g\in\mathcal{G}}$.

**Lemma A.2** (**Translational Nuisance** $\rightarrow_d$ **DCN Convolution**). *The MaxOut template matching and pooling operation (from Proposition A.1) for a set of translational nuisance variables $\mathcal{G} \equiv \mathcal{T}$ reduces to the traditional DCN convolution and max-pooling operation.*

*Proof.* Let the activation for a single output unit be $y_c(I)$. Then we have

$$y_c(I) \equiv \max_{g \in \mathcal{G}} \{w_{cg} \star_{LC} I\}$$

$$= \max_{t \in \mathcal{T}} \{\langle w_{ct}|I \rangle\}$$

$$= \max_{t \in \mathcal{T}} \{\langle T_t w_c|I \rangle\}$$

$$= \max_{t \in \mathcal{T}} \{\langle w_c|T_{-t}I \rangle\}$$

$$= \max_{t \in \mathcal{T}} \{(w_c \star_{\text{DCN}} I)_t\}$$

$$= \text{MaxPool}(w_c \star_{\text{DCN}} I).$$

where $\star_{\text{DCN}}$ is the traditional DCN Convolution operator. Finally, vectorizing in $c$ gives us the desired result $y(I) = \text{MaxPool}(\mathcal{W} \star_{\text{DCN}} I)$. $\qquad\square$

**Proposition A.3** (Max Pooling DCNs with ReLu Activations). *The discriminative relaxation of a noise-free GRMM **with translational nuisances and random missing data** is a single convolutional layer of a traditional DCN. The layer consists of a generalized convolution operation, followed by a ReLu activation function and a Max-Pooling operation.*

*Proof.* We will model completely random missing data as a nuisance transformation $a \in \mathcal{A} \equiv \{\text{keep}, \text{drop}\}$, where $a = \text{keep} = 1$ leaves the rendered image data untouched, while $a = \text{drop} = 0$ throws out the entire image after rendering. Thus, the switching variable $a$ models missing data. Critically, whether the data is missing is assumed to be *completely random* and thus independent of any other task variables, including the measurements (i.e. the image itself). Since the missingness of the evidence is just another nuisance, we can invoke Proposition A.1 to conclude that the discriminative relaxation of a noise-free GRMM with random missing data is also a MaxOut-DCN, but with a specialized structure which we now derive.

Mathematically, we decompose the nuisance variable $g \in \mathcal{G}$ into two parts $g = (t, a) \in \mathcal{G} = \mathcal{T} \times \mathcal{A}$, and then, following a similar line of reasoning as in Proposition A.1, we have

$$\hat{c}(I) = \underset{c \in \mathcal{C}}{\text{argmax}} \max_{g \in \mathcal{G}} p(c, g|I)$$

$$= \underset{c \in \mathcal{C}}{\text{argmax}} \left\{ \max_{g \in \mathcal{G}} \{w_{cg} \star_{LC} I\} \right\}$$

$$\overset{(a)}{=} \underset{c \in \mathcal{C}}{\text{argmax}} \left\{ \max_{t \in \mathcal{T}} \max_{a \in \mathcal{A}} \{a(\langle w_{ct}|I \rangle + b_{ct}) + b'_{ct} + b_a + b'_I\} \right\}$$

$$\overset{(b)}{=} \underset{c \in \mathcal{C}}{\text{argmax}} \left\{ \max_{t \in \mathcal{T}} \{\max\{(w_c \star_{\text{DCN}} I)_t, 0\} + b'_{ct} + b'_{\text{drop}} + b'_I\} \right\}$$

$$\overset{(c)}{=} \underset{c \in \mathcal{C}}{\text{argmax}} \left\{ \max_{t \in \mathcal{T}} \{\max\{(w_c \star_{\text{DCN}} I)_t, 0\} + b'_{ct}\} \right\}$$

$$\overset{(d)}{=} \underset{c \in \mathcal{C}}{\text{argmax}} \left\{ \max_{t \in \mathcal{T}} \{\max\{(w_c \star_{\text{DCN}} I)_t, 0\}\} \right\}$$

$$= \text{Choose} \{\text{MaxPool}(\text{ReLu}(\text{DCNConv}(I)))\}$$

$$= \text{DCN}(I; \theta).$$

In line (a) we calculated the log-posterior (ignoring $(c, g)$-independent constants)

$$\ln p(c, g|I) = \ln p(c, t, a|I)$$

$$= \ln p(I|c, t, a) + \ln p(c, t, a) + \ln p(I)$$

$$= \frac{1}{\sigma_x^2} \langle a\mu_{ct}|I \rangle - \frac{1}{2\sigma_x^2} (\|a\mu_{ct}\|_2^2 + \|I\|_2^2) + \ln p(c, t, a)$$

$$\equiv a(\langle w_{ct}|I \rangle + b_{ct}) + b'_{ct} + b_a + b'_I,$$

where $a \in \{0, 1\}$, $w_{ct} \equiv \frac{1}{\sigma_x^2} \mu_{ct}$, $b_{ct} \equiv -\frac{1}{2\sigma_x^2} \|\mu_{ct}\|_2^2$, $b_a \equiv \ln p(a)$, $b'_{ct} \equiv \ln p(c, t)$, $b'_I \equiv -\frac{1}{2\sigma_x^2} \|I\|_2^2$. In line (b), we use Lemma A.2 to write the expression in terms of the DCN convolution

operator, after which we invoke the identity $\max\{u,v\} = \max\{u-v, 0\} + v \equiv \mathrm{ReLu}(u-v) + v$ for real numbers $u, v \in \mathbb{R}$. Here we've defined $b'_{\mathrm{drop}} \equiv \ln p(a = \mathrm{drop})$ and we've used a slightly modified DCN convolution operator $\star_{\mathrm{DCN}}$ defined by $w_{ct} \star_{\mathrm{DCN}} I \equiv w_{ct} \star I + b_{ct} + \ln\left(\frac{p(a=\mathrm{keep})}{p(a=\mathrm{drop})}\right)$. Also, we observe that all the primed constants are independent of $a$ and so can be pulled outside of the $\max_a$. In line(c), the two primed constants that are also independent of $c, t$ can be dropped due to the $\mathrm{argmax}_{ct}$. Finally, in line (d), we assume a uniform prior over $c, t$. The resulting sequence of operations corresponds *exactly* to those applied in a single convolutional layer of a traditional DCN. □

## B   From the Deep Rendering Mixture Model to DCNs

Here we define the DRMM in full detail.

**Definition B.1** (**Deep Rendering Mixture Model (DRMM)**). *The Deep Rendering Mixture Model (DRMM) is a deep Gaussian Mixture Model (GMM) with special constraints on the latent variables. Generation in the DRMM takes the form:*

$$
\begin{aligned}
c^{(L)} &\sim Cat(\{\pi_{c^{(L)}}\}) \\
g^{(\ell)} &\sim Cat(\{\pi_{g^{(\ell)}}\}) \; \forall \ell \in [L] \equiv \{1, 2, \dots, L\} \\
\mu_{c^{(L)}g} &\equiv \Lambda_g \mu_{c^{(L)}} \\
&\equiv \Lambda_{g^{(1)}}^{(1)} \Lambda_{g^{(2)}}^{(2)} \dots \Lambda_{g^{(L-1)}}^{(L-1)} \Lambda_{g^{(L)}}^{(L)} \mu_{c^{(L)}} \\
I &\sim \mathcal{N}(\mu_{c^{(L)}g}, \Psi) \\
&= \mathcal{N}(\mu_{c^{(L)}g}, \sigma^2 \boldsymbol{I}_{D^{(0)}})
\end{aligned}
$$

*where the latent variables, parameters, and helper variables are defined as*

$$
\begin{aligned}
g^{(\ell)} &\equiv \left(g_{x^{(\ell)}}^{(\ell)}\right)_{x^{(\ell)} \in \mathcal{X}^{(\ell)}} \\
t^{(\ell)} &\equiv \left(t_{x^{(\ell)}}^{(\ell)}\right)_{x^{(\ell)} \in \mathcal{X}^{(\ell)}}, \; a^{(\ell)} \equiv \left(a_{x^{(\ell)}}^{(\ell)}\right)_{x^{(\ell)} \in \mathcal{X}^{(\ell)}} \\
g_{x^{(\ell)}}^{(\ell)} &\equiv \left(t_{x^{(\ell)}}^{(\ell)}, a_{x^{(\ell)}}^{(\ell)}\right) \\
t_{x^{(\ell)}}^{(\ell)} &\in \{UL, UR, LL, LR\} \\
a_{x^{(\ell)}}^{(\ell)} &\in \{0, 1\} \equiv \{OFF, ON\} \\
x^{(\ell)} &\in \mathcal{X}^{(\ell)} \equiv \{pixels \; in \; level \; \ell\} \in \mathbb{R}^{D^{(\ell)}} \\
\Lambda_{g^{(\ell)}}^{(\ell)} &= \Lambda_{t^{(\ell)}, a^{(\ell)}}^{(\ell)} \in \mathbb{R}^{D^{(\ell-1)} \times D^{(\ell)}} \\
&= T_{t^{(\ell)}}^{(\ell)} Z^{(\ell)} \Gamma^{(\ell)} M_{a^{(\ell)}}^{(\ell)} \\
M_{a^{(\ell)}}^{(\ell)} &\equiv diag\left(a^{(\ell)}\right) \in \mathbb{R}^{D^{(\ell)} \times D^{(\ell)}} \\
T_{t^{(\ell)}}^{(\ell)} &\equiv translation \; operator \; to \; position \; t^{(\ell)} \in \mathbb{R}^{D^{(\ell-1)} \times D^{(\ell-1)}} \\
Z^{(\ell)} &\equiv zero\text{-}padding \; operator \in \mathbb{R}^{D^{(\ell-1)} \times F^{(\ell)}} \\
\Gamma^{(\ell)} &\equiv \underset{x^{(\ell)} \in \mathcal{X}^{(\ell)}}{\otimes} \underbrace{\Gamma_{x^{(\ell)}}^{(\ell)}}_{F^{(\ell)} \times 1} \in \mathbb{R}^{F^{(\ell)} \times D^{(\ell)}} \\
\Gamma_{x^{(\ell)}}^{(\ell)} &\equiv \{filter \; bank \; at \; level \; \ell\} \in \mathbb{R}^{F^{(\ell)}} \\
F^{(\ell)} &\equiv W^{(\ell)} H^{(\ell)} C^{(\ell)} \\
&= \; size \; of \; the \; core \; templates \; at \; layer \; (\ell)
\end{aligned}
$$

For simplicity, in the following sections, we will use $c$ and $c^{(L)}$ interchangeably.

**Definition B.2** (**Nonnegative Deep Rendering Mixture Model (NN-DRM)**). *The Nonnegative Deep Rendering Mixture Model is defined as a DRMM (Definition B.1) with additional nonnegativity constraint(s) on the intermediate latent variables (rendered templates):*

$$z_n^{(\ell)} = \Lambda_{g_n^{(\ell+1)}} \cdots \Lambda_{g_n^{(L)}} \mu_{c_n^{(L)}} \geq 0 \quad \forall \ell \in \{1, \ldots, L\} \tag{9}$$

Following the same line of reasoning as in the main text, we will derive the Hard EM algorithm for the DRMM model.

## B.1 E-step: Computing the Soft Responsibilities

$$
\begin{aligned}
\gamma_{ncg} &\equiv p(c, g | I_n) \\
&= \frac{p(I_n | c, g; \theta) p(c, g | \theta)}{\sum_{c,g} p(I_n | c, g; \theta) p(c, g | \theta)} \\
&= \frac{\pi_{cg} |\Psi|^{-1/2} \exp\left(-\frac{1}{2} \|I_n - \mu_{cg}\|_{\Psi^{-1}}^2\right)}{Z},
\end{aligned}
$$

where the partition function $Z$ is defined as

$$Z(\theta) \equiv \sum_{c,g} \pi_{cg} |\Psi|^{-1/2} \exp\left(-\frac{1}{2} \|I_n - \mu_{cg}\|_{\Psi^{-1}}^2\right).$$

Since the numerator and denominator both contain $|\Psi|^{-1/2}$, the responsibilities simplify to

$$\gamma_{ncg} = \frac{\pi_{cg} \exp\left(-\frac{1}{2} \|I_n - \mu_{cg}\|_{\Psi^{-1}}^2\right)}{Z'}, \tag{10}$$

where $Z'$ is defined as

$$Z'(\theta) \equiv \sum_{cg} \pi_{cg} \exp\left(-\frac{1}{2} \|I_n - \mu_{cg}\|_{\Psi^{-1}}^2\right).$$

## B.2 E-step: Computing the Hard Responsibilities

Assuming isotropic noise $\Psi = \sigma^2 1_D$ and taking the zero-noise limit $\sigma^2 \to 0$, the term in the denominator $Z'(\theta)$ for which $\|I_n - \mu_{cg}\|_2^2$ is smallest will go to zero most slowly. Hence the responsibilities $\gamma_{ncg}$ will all approach zero, except for one term $(c^*, g^*)$, for which the $\gamma_{nc^*g^*}$ will approach one. [2] Thus, the soft responsibilities become hard responsibilities in the zero-noise limit:

$$\gamma_{ncg} \xrightarrow{\sigma \to 0} r_{ncg} \equiv \begin{cases} 1, & \text{if } (c, g) = \operatorname{argmax}_{c'g'} -\frac{1}{2} \|I_n - \mu_{c'g'}\|_2^2 \\ 0, & \text{otherwise} \end{cases} \tag{11}$$

## B.3 Useful Lemmas

In order to derive the E-step for the DRMM, we will need a few simple theoretical results. We prove them here.

**Definition B.3** (**Masking Operator**). *Let $a \in \{0, 1\}^d$ be a binary vector (mask) and let $\Lambda \in \mathbb{R}^{D \times d}$ be a real matrix. Then the **masking operator** $\mathcal{M}_a(\Lambda) \in \mathbb{R}^{D \times d}$ is defined as*

$$\mathcal{M}_a(\Lambda) \equiv \Lambda \cdot M_a \equiv \Lambda \cdot \operatorname{diag}(a),$$

*where $M_a \equiv \operatorname{diag}(a) \in \mathbb{R}^{d \times d}$ is the diagonal masking matrix.*

**Lemma B.4.** *The action of a masking operator on a vector $z \in \mathbb{R}^d$ can be written in several equivalent ways:*

$$
\begin{aligned}
\mathcal{M}_a(\Lambda)z &= \Lambda \cdot \text{diag}(a) \cdot z \\
&= \Lambda \cdot \text{diag}(a) \cdot \text{diag}(a) \cdot z \\
&= \Lambda[:, a] \cdot z[a] \\
&= \Lambda(a \odot z).
\end{aligned}
$$

*Here $\odot$ denotes the elementwise (Hadamard) product between two vectors and $\Lambda[:, a]$ is numpy notation for the subset of columns $\{j \in [D] : a_j = 1\}$ of $\Lambda$.*

*Proof.* The first equality is by definition. The second equality is a result of $a$ being binary since $a_i^2 = a_i$ for $a_i \in \{0, 1\}$. The third and fourth equalities result from the associativity of matrix multiplication. □

**Lemma B.5** (**Optimization with Masking Operators**). *Let $z, u \in \mathbb{R}^{D \times 1}$. Consider the optimization problem*

$$
\max_{a \in \{0,1\}^D} \mathcal{M}_a(z^T)u = \max_{a \in \{0,1\}^D} z^T M_a u \tag{12}
$$

*where $M_a \equiv \text{diag}(a)$. Then the optimization can be solved in closed form as:*

*(a)* $\displaystyle \max_{a \in \{0,1\}^D} \mathcal{M}_a(z^T)u = \mathbf{1}_D^T \text{ReLu}(z \odot u).$

*(b)* $\hat{a} \equiv \displaystyle \operatorname*{argmax}_{a \in \{0,1\}^D} \mathcal{M}_a(z^T)u = [z \odot u > 0] \in \{0, 1\}^D.$

*(c)* $M_{\hat{a}}u = \text{sgn}(z) \odot \text{ReLu}\left(\text{sgn}(z) \odot u\right).$

*(d) If $z \geq 0$, then $\hat{a} \equiv \displaystyle \operatorname*{argmax}_{a \in \{0,1\}^D} \mathcal{M}_a(z^T)u = [u > 0] \in \{0, 1\}^D$ is a maximizer, for which $M_{\hat{a}}u = \text{ReLu}(u).$*

*Proof.* *(a)* The maximum value can be computed as

$$
\begin{aligned}
v^\star &\equiv \max_{a \in \{0,1\}^D} \mathcal{M}_a(z^T)u \\
&= \max_{a \in \{0,1\}^D} z^T \text{diag}(a)u \\
&= \max_{a \in \{0,1\}^D} \sum_{i \in [D]} z_i a_i u_i \\
&= \sum_{i \in [D]} \max_{a_i \in \{0,1\}} a_i(z_i u_i) \\
&\equiv \sum_{i \in [D]} \hat{a}_i(z_i u_i) \\
&= \sum_{i \in [D]} [z_i u_i > 0] \cdot z_i u_i \\
&= \sum_{i \in [D]} \text{ReLu}(z_i u_i) \\
&= \mathbf{1}_D^T \text{ReLu}(z \odot u).
\end{aligned}
$$

*(b)* In the 4th line the vector optimization decouples into a set of independent scalar optimizations $\max_{a_i \in \{0,1\}} a_i(z_i u_i)$, each of which is solvable in closed form: $\hat{a}_i \equiv [z_i u_i > 0]$. Hence, the optimal

solution $\hat{a}$ is given by $\hat{a} = [z \odot u > 0]$.

*(c)* Substituting in $\hat{a}$, we get

$$
\begin{aligned}
\mathcal{M}_{\hat{a}} u &= u \odot [z \odot u > 0] \\
&= \underbrace{(\operatorname{sgn}(z) \odot \operatorname{sgn}(z))}_{\mathbf{1}_D} \odot u \odot [\operatorname{sgn}(z) \odot u > 0] \\
&= \operatorname{sgn}(z) \odot (\operatorname{sgn}(z) \odot u) \odot [\operatorname{sgn}(z) \odot u > 0] \\
&= \operatorname{sgn}(z) \odot \operatorname{ReLu}(\operatorname{sgn}(z) \odot u),
\end{aligned}
$$

where in the third and fourth equalities we have used the associativity of elementwise multiplication and the definition of ReLu, respectively.

*(d)* If $z \geq 0$, then when $z_i > 0$, $\hat{a}_i = [u_i > 0]$, and when $z_i = 0$, $\hat{a}_i$ can be either $0$ or $1$ since then $\max_{a_i \in \{0,1\}} a_i(z_i u_i) = 0 \, \forall a_i \in \{0, 1\}$. Therefore, if $z \geq 0$, $\hat{a} = [u > 0]$ is a solution of the optimization 12. It follows that $M_{\hat{a}} u = [u > 0]u = \operatorname{ReLu}(u)$.

$\square$

**Lemma B.6** (**Optimization with "Row" Max-Marginal**). *Let $z, u \in \mathbb{R}^{D \times 1}$. Consider the optimization problem*

$$
\max_{t \in \mathcal{T}^D} z^T u(t) = \max_{t \in \mathcal{T}^D} \sum_x z_x u(t)_x \tag{13}
$$

*where $\mathcal{T}$ is the set of possible **fine-scale** translations at location $x$. Also,*

$$
t \equiv \begin{bmatrix} \vdots \\ t_x \\ \vdots \end{bmatrix} \quad \text{and} \quad u(t) \equiv \begin{bmatrix} \vdots \\ u_x(t_x) \\ \vdots \end{bmatrix} \tag{14}
$$

*Then the optimization can be solved as:*

*(a)* $\displaystyle \max_{t \in \mathcal{T}^D} z^T u(t) = \sum_x |z_x| \max_{t_x \in \mathcal{T}} \operatorname{sgn}(z_x) u_x(t_x)$

*(b)* $\displaystyle \hat{t} = \operatorname*{argmax}_{t \in \mathcal{T}^D} z^T u(t) = \operatorname*{argmax}_t \operatorname{sgn}(z) \odot u(t) = \begin{bmatrix} \vdots \\ \operatorname*{argmax}_{t_x} \operatorname{sgn}(z_x) u_x(t_x) \\ \vdots \end{bmatrix}$

*(c)* $\displaystyle u(\hat{t}) = \operatorname{sgn}(z) \odot \max_t (\operatorname{sgn}(z) \odot u(t)) = \begin{bmatrix} \vdots \\ \operatorname{sgn}(z_x) \max_{t_x} \operatorname{sgn}(z_x) u_x(t_x) \\ \vdots \end{bmatrix}$

*(d) If $z \geq 0$, then $\hat{t} = \operatorname*{argmax}_t u(t) = \begin{bmatrix} \vdots \\ \operatorname*{argmax}_{t_x} u_x(t_x) \\ \vdots \end{bmatrix}$ is a maximizer for which $u(\hat{t}) = $*

$$
\max_t (u(t)) = \begin{bmatrix} \vdots \\ \max_{t_x} u_x(t_x) \\ \vdots \end{bmatrix}
$$

*Proof. (a)* The maximum value can be computed as

$$v^\star \equiv \max_{\{t_x \in \mathcal{T}\}_{x=1}^D} \sum_x z_x u_x(t_x)$$

$$= \sum_x \max_{t_x \in \mathcal{T}} z_x u_x(t_x)$$

$$= \sum_x \max_{t_x \in \mathcal{T}} |z_x| \operatorname{sgn}(z_x) u_x(t_x)$$

$$= \sum_x |z_x| \max_{t_x \in \mathcal{T}} \operatorname{sgn}(z_x) u_x(t_x)$$

*(b)* In the 2nd line the vector optimization decouples into a set of independent scalar optimizations $\max_{t_x \in \mathcal{T}} z_x u_x(t_x)$, each of which has the solution as follows: $\operatorname*{argmax}_{t_x} \operatorname{sgn}(z_x) u_x(t_x)$. Hence, the

optimal solution $\hat{t} = \operatorname*{argmax}_{t \in \mathcal{T}^D} z^T u(t) = \begin{bmatrix} \vdots \\ \operatorname*{argmax}_{t_x} \operatorname{sgn}(z_x) u_x(t_x) \\ \vdots \end{bmatrix} = \operatorname*{argmax}_t \operatorname{sgn}(z) \odot u(t)$.

*(c)* Substituting in $\hat{t}$, we obtain

$$v^\star = \sum_x |z_x| \operatorname{sgn}(z_x) u_x(\hat{t}_x)$$

$$= \sum_x z_x \operatorname{sgn}(z_x) (\operatorname{sgn}(z_x) u_x(\hat{t}_x))$$

$$= \sum_x z_x \operatorname{sgn}(z_x) \max_{t_x} \operatorname{sgn}(z_x) u_x(t_x)$$

$$= z^T \begin{bmatrix} \vdots \\ \operatorname{sgn}(z_x) \max_{t_x} \operatorname{sgn}(z_x) u_x(t_x) \\ \vdots \end{bmatrix}$$

Hence,

$$u(\hat{t}) = \begin{bmatrix} \vdots \\ \operatorname{sgn}(z_x) \max_{t_x} \operatorname{sgn}(z_x) u_x(t_x) \\ \vdots \end{bmatrix} = \operatorname{sgn}(z) \odot \max_t (\operatorname{sgn}(z) \odot u(t)),$$

*(d)* If $z \geq 0$, then when $z_i > 0$, $\hat{t}_x = \operatorname*{argmax}_{t_x} u_x(t_x)$, and when $z_i = 0$, $\hat{t}_x$ can take any value in its domain since then $\max_{t_x \in \mathcal{T}} \operatorname{sgn}(z_x) u_x(t_x) = 0 \, \forall t_x \in \mathcal{T}$. Therefore, if $z \geq 0$, $\hat{t}_x = \operatorname*{argmax}_{t_x} u_x(t_x)$ is a

solution of the optimization 13. It follows that $u(\hat{t}) = \max_t (u(t)) \equiv \begin{bmatrix} \vdots \\ \max_{t_x} u_x(t_x) \\ \vdots \end{bmatrix}$. $\square$

**Definition B.7 (Deep Masking Operator).** *Let $a^{(\ell)} \in \{0,1\}^{D^{(\ell)}}$ be a collection of binary (vector) masks and let $\Lambda^{(\ell)} \in \mathbb{R}^{D^{(\ell-1)} \times D^{(\ell)}}$ be a collection of (matrix) operators. Then the **deep masking operator** $\mathcal{M}_{\{a^{(\ell)}\}}(\{\Lambda^{(\ell)}\}) \in \mathbb{R}^{D^{(0)} \times D^{(L)}}$ is defined as*

$$\mathcal{M}_{\{a^{(\ell)}\}}(\{\Lambda^{(\ell)}\}) \equiv \prod_{\ell=1}^L \mathcal{M}_{a^{(\ell)}}(\Lambda^{(\ell)}) = \prod_{\ell=1}^L \Lambda^{(\ell)} \cdot M_{a^{(\ell)}},$$

*where $M_a \equiv \operatorname{diag}(a)$ is the diagonal masking matrix for mask $a$.*

## B.4 E-Step: Inference of Top-Level Category

**Theorem B.8** (**Inference in DRMM ⇒ Signed Convnets**). *Inference in the DRMM, according to the Dynamic Programming-based algorithm below, yields Signed DCNs. The inference algorithm has a bottom-up and top-down pass.*

*Proof.* Given input image $I_n \equiv I_n^{(0)}$, we infer $\hat{c}_n$ as follows:

$$\hat{c}_n = \operatorname*{argmax}_c \ \max_g -\frac{1}{2}\|I_n - \mu_{cg}\|_2^2$$

$$= \operatorname*{argmax}_c \ \max_g \mu_{cg}^T I_n - \frac{1}{2}\|I_n\|_2^2 - \frac{1}{2}\|\mu_{cg}\|_2^2$$

$$= \operatorname*{argmax}_c \ \max_g \mu_{cg}^T I_n - \frac{1}{2}\|\mu_{cg}\|_2^2,$$

where the last equality follow since $I_n$ is independent of $c, g$. We further assume that:

$$\alpha_{g^{(\ell)}} = 0 \ \forall \ell$$

$$\|\mu_{cg}\|_2^2 = \text{const} \ \forall c, g.$$

As a result, $\mu_{cg} = \Lambda_g \mu_c$ and the most probable class $\hat{c}_n$ is inferred as

$$\hat{c}_n = \operatorname*{argmax}_c \ \max_g \mu_{cg}^T I_n^{(0)} \tag{15}$$

$$= \operatorname*{argmax}_c \ \max_g (\Lambda_g \mu_c)^T I_n^{(0)} \tag{16}$$

$$= \operatorname*{argmax}_c \ \max_{g^{(L:1)}} \mu_c^T \Lambda_{g^{(L)}}^T \cdots \Lambda_{g^{(2)}}^T \Lambda_{g^{(1)}}^T I_n^{(0)} \tag{17}$$

$$= \operatorname*{argmax}_c \ \max_{g^{(L:2)}} \max_{t^{(1)}} \max_{a^{(1)}} \mu_c^T \Lambda_{g^{(L)}}^T \cdots \Lambda_{g^{(2)}}^T (M_{a^{(1)}} \Lambda_{t^{(1)}}^T) I_n^{(0)} \tag{18}$$

$$= \operatorname*{argmax}_c \ \max_{g^{(L:2)}} \max_{t^{(1)}} \max_{a^{(1)}} \underbrace{\left(\mu_c^T \Lambda_{g^{(L)}}^T \cdots \Lambda_{g^{(2)}}^T\right)}_{\equiv z^{(1)\downarrow T}} M_{a^{(1)}} \underbrace{\left(\Lambda_{t^{(1)}}^T I_n^{(0)}\right)}_{\equiv u_n^{(1)\uparrow}(t^{(1)})} \tag{19}$$

$$= \operatorname*{argmax}_c \ \max_{g^{(L:2)}} \max_{t^{(1)}} \max_{a^{(1)}} z^{(1)\downarrow T} M_{a^{(1)}} u_n^{(1)\uparrow}(t^{(1)}) \tag{20}$$

$$\overset{(a)}{=} \operatorname*{argmax}_c \ \max_{g^{(L:2)}} \max_{t^{(1)}} z^{(1)\downarrow T} M_{\hat{a}_n^{(1)}} u_n^{(1)\uparrow}(t^{(1)}) \tag{21}$$

$$\overset{(b)}{=} \operatorname*{argmax}_c \ \max_{g^{(L:2)}} z^{(1)\downarrow T} \left(s^{(1)\downarrow} \odot \max_{t^{(1)}} s^{(1)\downarrow} \odot \left(M_{\hat{a}_n^{(1)}} u_n^{(1)\uparrow}(t^{(1)})\right)\right) \tag{22}$$

$$\overset{(c)}{=} \operatorname*{argmax}_c \ \max_{g^{(L:2)}} z^{(1)\downarrow T} \left(s^{(1)\downarrow} \odot \max_{t^{(1)}} s^{(1)\downarrow} \odot \left(s^{(1)\downarrow} \odot \operatorname{ReLu}\left(s^{(1)\downarrow} \odot u_n^{(1)\uparrow}(t^{(1)})\right)\right)\right) \tag{23}$$

$$\overset{(d)}{=} \operatorname*{argmax}_c \ \max_{g^{(L:2)}} z^{(1)\downarrow T} \underbrace{\left(s^{(1)\downarrow} \odot \operatorname{MaxPool}\left(\operatorname{ReLu}\left(\operatorname{diag}(s^{(1)\downarrow}) u_n^{(1)\uparrow}(\mathcal{T})\right)\right)\right)}_{\equiv I_n^{(1)}(s^{(1)\downarrow})} \tag{24}$$

$$= \operatorname*{argmax}_c \ \max_{g^{(L:2)}} \mu_c^T \Lambda_{g^{(L)}}^T \cdots \Lambda_{g^{(2)}}^T I_n^{(1)} \tag{25}$$

In line (a), we employ Lemma B.5*(b)* to infer the optimal $\hat{a}_n^{(1)}$. In line (b) and (c), we employ B.6*(c)* and Lemma B.5*(c)* to calculate the max-product message $I_n^{(1)}$ to be sent to the next layer. Notice that here $s^{(1)\downarrow} = \operatorname{sgn}\left(z^{(1)\downarrow}\right)$. In line (b), $\hat{t}_n^{(1)}$ is implicitly inferred via Lemma B.6*(b)*. In line (d), $s^{(1)\downarrow} \odot s^{(1)\downarrow}$ becomes a vector of all 1's. Also, in the same line, $\operatorname{diag}(s^{(1)\downarrow})$ is a diagonal matrix with diagonal $s^{(1)\downarrow}$ and $u_n^{(1)\uparrow}(\mathcal{T})$ is a matrix $[u_{nxt}]$ where rows are indexed by $x \in \mathcal{X}$ and columns by $t \in \mathcal{T}$. It corresponds to the output of the convolutional layer in a DCN, prior to the ReLu and spatial max-pooling operators.

Note that we have succeeded in expressing the optimization (Eq. 17) recursively in terms of a one level smaller sub-problem (Eq. 25). Iterating this procedure yields a set of recurrence relations, which define our *Dynamic Programming (DP)* algorithm for the bottom-up and top-down inference in the DRMM:

**Bottom-Up E-Step (E$_\uparrow$):**

$$u_n^{(\ell)\uparrow} = \Lambda_{t^{(\ell)}}^T I_n^{(\ell-1)} \tag{26}$$

$$s^{(\ell)\downarrow} = \mathrm{sgn}\left(z^{(\ell)\downarrow}\right) \tag{27}$$

$$\forall s^{(\ell)\downarrow} \in \{\pm 1\}^{D^{(\ell)}} : \hat{a}_n^{(\ell)\updownarrow}(s^{(\ell)\downarrow}) = [s^{(\ell)\downarrow} \odot u_n^{(\ell)\uparrow} > 0] \tag{28}$$

$$\forall s^{(\ell)\downarrow} \in \{\pm 1\}^{D^{(\ell)}} : \hat{t}_n^{(\ell)\updownarrow}(s^{(\ell)\downarrow}) = \underset{t^{(\ell)}}{\mathrm{argmax}}\ s^{(\ell)\downarrow} \odot u_n^{(\ell)\uparrow}(t^{(\ell)}) \tag{29}$$

$$I_n^{(\ell)}(s^{(\ell)\downarrow}) = M_{\hat{a}_n^{(\ell)}}\left(\Lambda_{\hat{t}^{(\ell)}}^T I_n^{(\ell-1)}\right) \tag{30}$$

$$= s^{(\ell)\downarrow} \odot \mathrm{MaxPool}\left(\mathrm{ReLu}\left(\mathrm{diag}(s^{(1)\downarrow})u_n^{(1)\uparrow}(\mathcal{T})\right)\right) \tag{31}$$

$$\hat{c}_n^{(L)} = \underset{c^{(L)}}{\mathrm{argmax}}\ \mu_{c^{(L)}}^T I_n^{(L)} \tag{32}$$

**Top-Down/Traceback E-Step (E$_\uparrow$):**

$$\hat{z}_n^{(\ell)\downarrow} = \Lambda_{\hat{g}_n^{(\ell+1)}} \cdots \Lambda_{\hat{g}_n^{(L)}} \mu_{\hat{c}_n^{(L)}} \tag{33}$$

$$= \Lambda_{\hat{g}_n^{(\ell+1)}} \hat{z}_n^{(\ell+1)\downarrow} \tag{34}$$

$$\hat{s}_n^{(\ell)\downarrow} = \mathrm{sgn}(\hat{z}_n^{(\ell)\downarrow}) \tag{35}$$

$$\hat{a}_n^{(\ell)\updownarrow} = \hat{a}_n^{(\ell)\updownarrow}(\hat{s}_n^{(\ell)\downarrow}) = [\hat{s}_n^{(\ell)\downarrow} \odot u_n^{(\ell)\uparrow} > 0] \tag{36}$$

$$\hat{t}_n^{(\ell)\updownarrow} = \hat{t}_n^{(\ell)\updownarrow}(s_n^{(\ell)\downarrow}) = \underset{t^{(\ell)}}{\mathrm{argmax}}\ s_n^{(\ell)\downarrow} \odot u_n^{(\ell)\uparrow}(t^{(\ell)}) \tag{37}$$

where $u_n^{(\ell)\uparrow}$ and $\hat{z}_n^{(\ell)\downarrow}$ are the bottom-up and top-down net inputs into layer $\ell$, respectively. $\qquad\square$

**Corollary B.9** (**Inference in the NN-DRMM $\Rightarrow$ Convnets**). *Inference in the NN-DRMM according to the Dynamic Programming-based algorithm above yields ReLu DCNs.*

*Proof.* The NN-DRMM assumes that the intermediate rendered latent variables $z_n^{(\ell)} \geq 0$ for all $\ell$, which implies that the signs are also nonnegative i.e., $s_n^{(\ell)} \geq 0$. This in turn, according to Lemma B.5*(d)* and B.6*(d)*, reduces Eqs. 31, 32, 36 and 37 to

$$\mathrm{E}_\uparrow : I_n^{(\ell)} = \mathrm{MaxPool}\,\mathrm{ReLu}\left(u_n^{(\ell)\uparrow}\right) \tag{38}$$

$$\hat{c}_n^{(L)} = \underset{c^{(L)}}{\mathrm{argmax}}\ \mu_{c^{(L)}}^T I_n^{(L)} \tag{39}$$

$$\mathrm{E}_\downarrow : \hat{a}_n^{(\ell)} = [u_n^{(\ell)\uparrow} > 0] \tag{40}$$

$$\hat{t}_n^{(\ell)} = \underset{t^{(\ell)}}{\mathrm{argmax}}\ u_n^{(\ell)\uparrow}(t^{(\ell)}), \tag{41}$$

which is equivalent to feedforward propagation in a DCN. Note that the the top-down step no longer requires information from the deeper levels, and so it can be computed in the bottom-up step instead. $\qquad\square$

**Remark:** Note that the vector max notation $\max\limits_t u(t) = \begin{bmatrix} \vdots \\ \max\limits_{t_x} u_x(t_x) \\ \vdots \end{bmatrix}$ is the same as the max nota-

tion we use in our arXiv post. It refers to the **row** max-marginals of the matrix $u(t) \equiv [u_{xt}]_{x \in \mathcal{X}, t \in \mathcal{T}}$ with respect to latent variables $t$.

Figure 3: Neural network implementation of shallow Rendering Model EM algorithm.

## C Rendering Factor Model (RFM) Architecture

## D Transforming a Generative Classifier into a Discriminative One

Before we formally define the procedure, some preliminary definitions and remarks will be helpful. A generative classifier models the joint distribution $p(c, I)$ of the input features *and* the class labels. It can then classify inputs by using Bayes Rule to calculate $p(c|I) \propto p(c, I) = p(I|c)p(c)$ and picking the most likely label $c$. Training such a classifier is known as *generative learning*, since one can generate synthetic features $I$ by sampling the joint distribution $p(c, I)$. Therefore, a generative classifier learns an *indirect* map from input features $I$ to labels $c$ by modeling the joint distribution $p(c, I)$ of the labels *and* the features.

In contrast, a discriminative classifier parametrically models $p(c|I) = p(c|I; \theta_d)$ and then trains on a dataset of input-output pairs $\{(I_n, c_n)\}_{n=1}^{N}$ in order to estimate the parameter $\theta_d$. This is known as *discriminative learning*, since we directly discriminate between different labels $c$ given an input feature $I$. Therefore, a discriminative classifier learns a direct map from input features $I$ to labels $c$ by *directly* modeling the conditional distribution $p(c|I)$ of the labels *given* the features.

Given these definitions, we can now define the *discriminative relaxation* procedure for converting a generative classifier into a discriminative one. Starting with the standard learning objective for a generative classifier, we will employ a series of transformations and relaxations to obtain the learning

Figure 4: Graphical depiction of discriminative relaxation procedure. (A) The Rendering Model (RM) is depicted graphically, with mixing probability parameters $\pi_{cg}$ and rendered template parameters $\lambda_{cg}$. Intuitively, we can interpret the discriminative relaxation as a *brain-world transformation* applied to a generative model. According to this interpretation, instead of the world generating images and class labels (A), we instead imagine the world generating images $I_n$ via the rendering parameters $\tilde{\theta} \equiv \theta_{\text{world}}$ while the brain generates labels $c_n, g_n$ via the classifier parameters $\eta_{\text{dis}} \equiv \eta_{\text{brain}}$ (B). The brain-world transformation converts the RM (A) to an equivalent graphical model (B), where an extra set of parameters $\tilde{\theta}$ and constraints (arrows from $\theta$ to $\tilde{\theta}$ to $\eta$) have been introduced. Discriminatively relaxing these constraints (B, red X's) yields the single-layer DCN as the discriminative counterpart to the original generative RM classifier in (A).

objective for a discriminative classifier. Mathematically, we have

$$
\begin{aligned}
\max_{\theta} L_{\text{gen}}(\theta) &\equiv \max_{\theta} \sum_n \ln p(c_n, I_n | \theta) \\
&\overset{(a)}{=} \max_{\theta} \sum_n \ln p(c_n | I_n, \theta) + \ln p(I_n | \theta) \\
&\overset{(b)}{=} \max_{\theta, \tilde{\theta} : \theta = \tilde{\theta}} \sum_n \ln p(c_n | I_n, \theta) + \ln p(I_n | \tilde{\theta}) \\
&\overset{(c)}{\leq} \max_{\theta} \underbrace{\sum_n \ln p(c_n | I_n, \theta)}_{\equiv L_{\text{cond}}(\theta)} \\
&\overset{(d)}{=} \max_{\eta : \eta = \rho(\theta)} \sum_n \ln p(c_n | I_n, \eta) \\
&\overset{(e)}{\leq} \max_{\eta} \underbrace{\sum_n \ln p(c_n | I_n, \eta)}_{\equiv L_{\text{dis}}(\eta)},
\end{aligned}
\tag{42}
$$

where the $L$'s are the *generative*, *conditional* and *discriminative* log-likelihoods, respectively. In line (a), we used the Chain Rule of Probability. In line (b), we introduced an extra set of parameters $\tilde{\theta}$ while also introducing a constraint that enforces equality with the old set of generative parameters $\theta$. In line (c), we relax the equality constraint (first introduced by Bishop, LaSerre and Minka in [4]), allowing the classifier parameters $\theta$ to differ from the image generation parameters $\tilde{\theta}$. In line (d), we pass to the *natural parametrization* of the exponential family distribution $I|c$, where the natural parameters $\eta = \rho(\theta)$ are a fixed function of the conventional parameters $\theta$. This constraint on the natural parameters ensures that optimization of $L_{\text{cond}}(\eta)$ yields the same answer as optimization of $L_{\text{cond}}(\theta)$. And finally, in line (e) we relax the natural parameter constraint to get the learning objective for a discriminative classifier, where the parameters $\eta$ are now free to be optimized. A graphical model depiction of this process is shown in Fig. 4.

In summary, starting with a generative classifier with learning objective $L_{\text{gen}}(\theta)$, we complete steps (a) through (e) to arrive at a discriminative classifier with learning objective $L_{\text{dis}}(\eta)$. We refer to this process as a *discriminative relaxation of a generative classifier* and the resulting classifier is a *discriminative counterpart to the generative classifier*.

**goose**     **ostrich**     **limousine**

Figure 5: Results of activity maximization on the ImageNet dataset [23]. For a given class $c$, activity-maximizing inputs are superpositions of various poses of the object, with distinct patches $\mathcal{P}_i$ containing distinct poses $g^*_{\mathcal{P}_i}$, as predicted by Eq. 44. Figure adapted with permission from the authors.

# E  Derivation of Closed-Form Expression for Activity-Maximizing Images

Results of running activity maximization are shown in Fig. 5 for completeness. Mathematically, we seek the image $I$ that maximizes the score $S(c|I)$ of a specific object class. Using the DRM, we have

$$
\begin{aligned}
\max_I S(c^{(L)}|I) &= \max_I \max_{g \in \mathcal{G}} \langle \frac{1}{\sigma^2}\mu(c^{(L)}, g^{(\ell)})|I\rangle \\
&\propto \max_{g \in \mathcal{G}} \max_I \langle \mu(c^{(L)}, g)|I\rangle \\
&= \max_{g \in \mathcal{G}} \max_{I_{\mathcal{P}_1}} \cdots \max_{I_{\mathcal{P}_p}} \langle \mu(c^{(L)}, g)| \sum_{\mathcal{P}_i \in \mathcal{P}} I_{\mathcal{P}_i}\rangle \\
&= \max_{g \in \mathcal{G}} \sum_{\mathcal{P}_i \in \mathcal{P}} \max_{I_{\mathcal{P}_i}} \langle \mu(c^{(L)}, g)|I_{\mathcal{P}_i}\rangle \\
&= \max_{g \in \mathcal{G}} \sum_{\mathcal{P}_i \in \mathcal{P}} \langle \mu(c^{(L)}, g)|I^*_{\mathcal{P}_i}(c^{(L)}, g)\rangle \\
&= \sum_{\mathcal{P}_i \in \mathcal{P}} \langle \mu(c^{(L)}, g)|I^*_{\mathcal{P}_i}(c^{(L)}, g^*_{\mathcal{P}_i})\rangle,
\end{aligned}
\tag{43}
$$

where $I^*_{\mathcal{P}_i}(c^{(\ell)}, g) \equiv \operatorname{argmax}_{I_{\mathcal{P}_i}} \langle \mu(c^{(\ell)}, g)|I_{\mathcal{P}_i}\rangle$ and $g^*_{\mathcal{P}_i} = g^*(c^{(\ell)}, \mathcal{P}_i) \equiv \operatorname{argmax}_{g \in \mathcal{G}} \langle \mu(c^{(\ell)}, g)|I^*_{\mathcal{P}_i}(c^{(\ell)}, g)\rangle$. In the third line, the image $I$ is decomposed into $P$ patches $I_{\mathcal{P}_i}$ of the same size as $I$, with all pixels outside of the patch $\mathcal{P}_i$ set to zero. The $\max_{g \in \mathcal{G}}$ operator finds the most probable $g^*_{\mathcal{P}_i}$ within each patch. The solution $I^*$ of the activity maximization is then the sum of the individual activity-maximizing patches

$$
I^* \equiv \sum_{\mathcal{P}_i \in \mathcal{P}} I^*_{\mathcal{P}_i}(c^{(\ell)}, g^*_{\mathcal{P}_i}) \propto \sum_{\mathcal{P}_i \in \mathcal{P}} \mu(c^{(\ell)}, g^*_{\mathcal{P}_i}).
\tag{44}
$$

# F  From the DRMM to Decision Trees

In this section we show that, like DCNs, Random Decision Forests (RDFs) can also be derived from the DRMM model. Instead of translational and switching nuisances, we will show that an *additive mutation nuisance process* that generates a hierarchy of categories (e.g., evolution of a taxonomy of living organisms) is at the heart of the RDF.

## F.1  The Evolutionary Deep Rendering Mixture Model

We define the *Evolutionary DRMM* (E-DRMM) as a DRMM with an evolutionary tree of categories. Samples from the model are generated by starting from the root ancestor template and randomly

mutating the templates. Each child template is an additive "mutation" of its parent, where the specific mutation does not depend on the parent (see Eq.45 below). At the leaves of the tree, a sample is generated by adding Gaussian pixel noise. Like in the DRMM, given $c^{(L)} \sim \text{Cat}(\pi_{c^{(L)}})$ and $g^{(\ell+1)} \sim \text{Cat}(\pi_{g^{(\ell+1)}})$, with $c^{(L)} \in \mathcal{C}^L$ and $g^{(\ell+1)} \in \mathcal{G}^{\ell+1}$ where $\ell = 1, 2, \cdots, L$, the template $\mu_{c^{(L)}g}$ and the image $I$ are rendered as

$$\mu_{c^{(L)}g} = \Lambda_g \mu_{c^{(L)}} \equiv \Lambda_{g^{(1)}} \cdots \Lambda_{g^{(L)}} \cdot \mu_{c^{(L)}}$$

$$\equiv \mu_{c^{(L)}} + \alpha_{g^{(L)}} + \cdots + \alpha_{g^{(1)}}, \quad g = \{g^{(\ell)}\}_{\ell=1}^L$$

$$I \sim \mathcal{N}(\mu_{c^{(L)}g}, \sigma^2 1_D) \in \mathbb{R}^D.$$

Here, $\Lambda_{g^{(\ell)}}$ has a special structure due to the additive mutation process: $\Lambda_{g^{(\ell)}} = [\mathbf{1} \,|\, \alpha_{g^{(\ell)}}]$, where $\mathbf{1}$ is the identity matrix. The rendering path represents template evolution and is defined as the sequence $(c^{(L)}, g^{(L)}, \ldots, g^{(\ell)}, \ldots, g^{(1)})$ from the root ancestor template down to the individual pixels at $\ell = 0$. $\mu_{c^{(L)}}$ is an abstract template for the root ancestor $c^{(L)}$, and $\sum_\ell \alpha_{g^{(\ell)}}$ represents the sequence of local nuisance transformations, in this case, the accumulation of many additive mutations.

As with the DRMM, we can cast the E-DRMM into an incremental form by defining an intermediate class $c^{(\ell)} \equiv (c^{(L)}, g^{(L)}, \ldots, g^{(\ell+1)})$ that intuitively represents a partial evolutionary path up to level $\ell$. Then, the mutation from level $\ell + 1$ to $\ell$ can be written as

$$\mu_{c^{(\ell)}} = \Lambda_{g^{(\ell+1)}} \cdot \mu_{c^{(\ell+1)}} = \mu_{c^{(\ell+1)}} + \alpha_{g^{(\ell+1)}}. \tag{45}$$

Here, $\alpha_{g^{(\ell)}}$ is the mutation added to the template at level $\ell$ in the evolutionary tree.

### F.2 Inference with the E-DRM Yields a Decision Tree

Since the E-DRMM is an RMM with a hierarchical prior on the rendered templates, we can use Eq.3 to derive the E-DRMM inference algorithm for $\hat{c}^{(L)}(I)$ as:

$$\hat{c}^{(L)}(I) = \underset{c^{(L)} \in \mathcal{C}^L}{\text{argmax}} \max_{g \in \mathcal{G}} \langle \mu_{c^{(L)}} + \alpha_{g^{(L)}} + \cdots + \alpha_{g^{(1)}} | I \rangle$$

$$= \underset{c^{(L)} \in \mathcal{C}^L}{\text{argmax}} \max_{g^{(1)} \in \mathcal{G}^1} \cdots \max_{g^{(L-1)} \in \mathcal{G}^{L-1}} \langle \underbrace{\mu_{c^{(L)}} + \alpha_{g^{(L)*}}}_{\equiv \mu_{c^{(L-1)}}} + \cdots + \alpha_{g^{(1)}} | I \rangle$$

$$\cdots$$

$$\equiv \underset{c^{(L)} \in \mathcal{C}^L}{\text{argmax}} \langle \mu_{c^{(L)}g^*} | I \rangle. \tag{46}$$

where $\mu_{c^{(\ell)}}$ has been defined in the second line. Here, we assume that the sub-trees are well-separated. In the last lines, we repeatedly use the distributivity of max over sums, resulting in the iteration

$$g^*_{c^{(\ell+1)}} \equiv \underset{g^{(\ell+1)} \in \mathcal{G}^{\ell+1}}{\text{argmax}} \langle \underbrace{\mu_{c^{(\ell+1)}g^{(\ell+1)}}}_{\equiv W^{(\ell+1)}} | I \rangle$$

$$\equiv \text{ChooseChild}(\text{Filter}(I)). \tag{47}$$

Eqs.46 and 47 define a *Decision Tree*. The leaf label histograms at the end of a decision tree plays a similar role as the SoftMax regression layer in DCNs. Applying bagging [5] on decision trees yield a Random Decision Forest (RDF).

## G  Unifying the Probabilistic and Neural Network Perspectives

## H  Additional Experimental Results

### H.1  Learned Filters and Image Reconstructions

Filters and reconstructed images are shown in Fig. 6.

### H.2  Additional Training Results

More results plus comparison to other related work are given in Table 3.

Table 2: Summary of probabilistic and neural network perspectives for DCNs. The DRMM provides a probabilistic interpretation for all of the common elements of DCNs relating to the underlying model, inference algorithm, and learning rules.

| Aspect | Neural Nets Perspective *(Deep Convolutional Neural Networks)* | Probabilistic Perspective *(Deep Rendering Model)* |
|---|---|---|
| **Model** | Weights and biases of filters at a given layer | Partial Rendering at a given abstraction level/scale |
| | Number of Layers | Number of Abstraction Levels |
| | Number of Filters in a layer | Number of Clusters/Classes at a given abstraction level |
| | Implicit in network weights; can be computed by product of weights over all layers or by activity maximization | Category prototypes are finely detailed versions of coarser-scale super-category prototypes. Fine details are modeled with affine nuisance transformations. |
| **Inference** | Forward propagation thru DCN | Exact bottom-up inference via Max-Sum Message Passing (with Max-Product for Nuisance Factorization). |
| | Input and Output Feature Maps | Probabilistic Max-Sum Messages (real-valued functions of variables nodes) |
| | Template matching at a given layer (convolutional, locally or fully connected) | Local computation at factor node (log-likelihood of measurements) |
| | Max-Pooling over local pooling region | Max-Marginalization over Latent Translational Nuisance transformations |
| | Rectified Linear Unit (ReLU). Sparsifies output activations. | Max-Marginalization over Latent Switching state of Renderer. Low prior probability of being ON. |
| **Learning** | Stochastic Gradient Descent | Batch Discriminative EM Algorithm with Fine-to-Coarse E step + Gradient M-step. *No coarse-to-fine pass in E-step.* |
| | N/A | Full EM Algorithm |
| | Batch-Normalized SGD | Discriminative Approximation to Full EM (assumes Diagonal Pixel Covariance) |

Figure 6: (Left) Filters learned from 60,000 unlabeled MNIST samples and (Right) reconstructed images from the Shallow Rendering Mixture Model

# I   Model Configurations

In our experiments, configurations of the RFM and 2-layer DRFM are similar to LeNet5 [10] and its variants. Also, configurations of the 5-layer DRMM (for MNIST) and the 9-layer DRMM (for CIFAR10) are similar to Conv-Small and Conv-Large architectures in [26, 17], respectively.

Table 3: Test error (%) for supervised, unsupervised and semi-supervised training on MNIST using $N_U = 60K$ unlabeled images and $N_L \in \{100, 600, 1K, 3K, 60K\}$ labeled images.

| Model | Test Error (%) | | | | |
|---|---|---|---|---|---|
| | $N_L = 100$ | $N_L = 600$ | $N_L = 1K$ | $N_L = 3K$ | $N_L = 60K$ |
| RFM sup | - | - | - | - | 1.21 |
| Convnet 1-layer sup | - | - | - | - | 1.30 |
| DRMM 5-layer sup | - | - | - | - | 0.89 |
| Convnet 5-layer sup | - | - | - | - | **0.81** |
| RFM unsup-pretr | 16.2 | 5.65 | 4.64 | 2.95 | 1.17 |
| DRMM 5-layer unsup-pretr | **12.03** | **3.61** | **2.73** | **1.68** | **0.58** |
| SWWAE unsup-pretr [31] | - | 9.80 | 6.135 | 4.41 | - |
| RFM semi-sup | 14.47 | 5.61 | 4.67 | 2.96 | 1.27 |
| DRMM 5-layer semi-sup | 3.50 | **1.56** | 1.67 | **0.91** | 0.51 |
| Convnet [10] | 22.98 | 7.86 | 6.45 | 3.35 | - |
| TSVM [30] | 16.81 | 6.16 | 5.38 | 3.45 | - |
| CAE [19] | 13.47 | 6.3 | 4.77 | 3.22 | - |
| MTC [18] | 12.03 | 5.13 | 3.64 | 2.57 | - |
| PL-DAE [11] | 10.49 | 5.03 | 3.46 | 2.69 | - |
| WTA-AE [13] | - | 2.37 | 1.92 | - | - |
| SWWAE no dropout [31] | $9.17 \pm 0.11$ | $4.16 \pm 0.11$ | $3.39 \pm 0.01$ | $2.50 \pm 0.01$ | - |
| SWWAE with dropout [31] | $8.71 \pm 0.34$ | $3.31 \pm 0.40$ | $2.83 \pm 0.10$ | $2.10 \pm 0.22$ | - |
| M1+TSVM [8] | $11.82 \pm 0.25$ | 5.72 | 4.24 | 3.49 | - |
| M1+M2 [8] | $3.33 \pm 0.14$ | $2.59 \pm 0.05$ | $2.40 \pm 0.02$ | $2.18 \pm 0.04$ | - |
| Skip Deep Generative Model [12] | 1.32 | - | - | - | - |
| LadderNetwork [17] | $1.06 \pm 0.37$ | - | $0.84 \pm 0.08$ | - | - |
| Auxiliary Deep Generative Model [12] | 0.96 | - | - | - | - |
| ImprovedGAN [21] | $\mathbf{0.93 \pm 0.065}$ | - | - | - | - |
| catGAN [25] | $1.39 \pm 0.28$ | - | - | - | - |

Table 4: Test error rates (%) between 2-layer DRMM and 9-layer DRMM trained with semi-supervised EG and other best published results on CIFAR10 using $N_U = 50K$ unlabeled images and $N_L \in \{4K, 50K\}$ labeled images

| Model | $N_L = 4K$ | $N_L = 50K$ |
|---|---|---|
| Convnet [10] | 43.90 | 27.17 |
| Conv-Large [26] | - | 9.27 |
| CatGAN [25] | $19.58 \pm 0.46$ | 9.38 |
| ImprovedGAN [21] | $\mathbf{18.63 \pm 2.32}$ | - |
| LadderNetwork [17] | $20.40 \pm 0.47$ | - |
| DRMM 2-layer | 39.2 | 24.60 |
| DRMM 9-layer | 23.24 | 11.37 |

Table 5: Comparison of RFM, 2-layer DRMM and 5-layer DRMM against Stacked What-Where Auto-encoders with various regularization approaches on the MNIST dataset. $N$ is the number of labeled images used, and there is no extra unlabeled image.

| Model | $N = 100$ | $N = 600$ | $N = 1K$ | $N = 3K$ |
|---|---|---|---|---|
| SWWAE (3 layers) [31] | $\mathbf{10.66 \pm 0.55}$ | $4.35 \pm 0.30$ | $3.17 \pm 0.17$ | $2.13 \pm 0.10$ |
| SWWAE (3 layers) + dropout on convolution [31] | $14.23 \pm 0.94$ | $4.70 \pm 0.38$ | $3.37 \pm 0.11$ | $2.08 \pm 0.10$ |
| SWWAE (3 layers) + L1 [31] | $10.91 \pm 0.29$ | $4.61 \pm 0.28$ | $3.55 \pm 0.31$ | $2.67 \pm 0.25$ |
| SWWAE (3 layers) + noL2M [31] | $12.41 \pm 1.95$ | $4.63 \pm 0.24$ | $3.15 \pm 0.22$ | $2.08 \pm 0.18$ |
| Convnet (1 layer) | 18.33 | 10.36 | 8.07 | 4.47 |
| RFM (1 layer) | 22.68 | 6.51 | 4.66 | 3.55 |
| DRMM 2-layer | 12.56 | 6.50 | 4.75 | 2.66 |
| DRMM 5-layer | 11.97 | **3.70** | **2.72** | **1.60** |

## Footnotes

[2]Technically, there can be multiple maximizers and the algorithms below can be generalized to handle this case. But we focus on the case with just one unique maximum for simplicity.