[Reviews · NeurIPS 2016]

Reviewer 1

Summary

This draft proposed a probabilistic model for deep learning, motivated by the idea of factoring out nuisance variation in the input. The probabilistic model stacks hierarchical layers of mixture of factor analyzers, in order to model the correlations between pixels. Inference algorithm using expectation-maximization and messaging passing is also derived. Empirical results on MNIST shows the potential of the the proposed probabilistic framework.

Qualitative Assessment

The draft provided a fresh view on deep convolutional neural networks (CNN), with the following strength points: 1) The probabilistic framework is elegant and supported by earlier work on probabilistic mixture of factor analyzers and its deep versions. 2) The motivation of factoring out nuisance variation will also be inspiring to other researchers in the field. Using the same philosophical viewpoint, statistical models other than CNN could also be explained in a probabilistic and principled way. 3) The inference algorithm sheds light on the mechanics behind the convolution and max-pooling operators in CNN. It also has the advantage of characterizing uncertainty in the data and model. Nevertheless, I saw the following points that can be improved: 1) The explanation for the notations is too concise, and sometimes missing. For example, in equation (3) the reader can guess the notations for inner products, but it is better to explain it explicitly in the text. Another example is in Section 3.2. In the first equation of that section, a standard form of sparse factor analysis model is used. However, the binary factor a is missing in the following equations, leaving only z. This would cause some confusions to the reader. 2) The probabilistic framework is sound but the experiment part can be made more solid. Other standard image datasets (e.g., ImageNet, CIFAR) should be tested besides MNIST. Also, more experiments are needed to demonstrate the advantage of stacking deeper layers. 3) There are some earlier work on explaining CNN from the viewpoint of message passing, for example, the following one and the papers cited therein. It would be nice to compare and connect to their modeling approach: Zheng, Shuai, Sadeep Jayasumana, Bernardino Romera-Paredes, Vibhav Vineet, Zhizhong Su, Dalong Du, Chang Huang, and Philip HS Torr. "Conditional random fields as recurrent neural networks." In Proceedings of the IEEE International Conference on Computer Vision, pp. 1529-1537. 2015.

Confidence in this Review

2-Confident (read it all; understood it all reasonably well)


Reviewer 2

Summary

The authors present a generative model, called a Rendering Model which explicitly reasons about class and nuisance variation. More formally, their model generates pixels by patch and assumes a (gaussian) noisy image pixel, and an image template that is a function of the object class and a nuisance variable. The model also makes use of a switching variable that allows the model to deal with two objects overlapping (occlusion or transparency). They demonstrate that inference in the Rendering Model is (nearly) equivalent to a one-layer conv net which uses a batch-norm (in the beginning, rather than at the end), a fully connected or convolutional layer, a max pooling layer and a relu layer. They combine multiple Rendering Models to create a Deep Rendering Model (DRM). The DRM is a graphical model whose layers are modified rendering models resulting in a series of affine transformations that render finer and finer details of the object as the model progresses. They argue that a relaxed form of the DRM produces a convolutional neural network. The DRM can be trained via EM which they argue provides faster convergence than standard gradient descent. Given their model and the drawn equivalences to convolutional neural networks, they argue that the DRM provides several insights into convolutional networks strengths and weaknesses. Firstly, since DRM inference is equivalent to max-sum message passing, a conv net is effectively doing the same thing (making a number of assumptions about its structure). Secondly, they suggest that convnets poor performance on transparent objects is due to the ambiguity of the label being assigned to the transparency. They relate this weakness to the constraint in the DRM which enforces only a single object being rendered per pixel. Thirdly, they argue that like DRMs, convnets represent appearance models for classes in a factorized-mixture form over various levels of abstraction. They validate their method on MNIST and experiment with comparing their supervised model with a simple convnet as well as demonstrating decent results for unsupervised and semi-supervised variations of their model. They illustrate decent performances for both and very good performance using the supervised DRM.

Qualitative Assessment

The paper presents a very intriguing model and make an ambitious argument about its equivalence with convolutional networks. If anything, the paper suffers slightly from an abundance of content which was very difficult (I imagine) to squeeze into the allotted space. The model is sure to be of interest to anybody working with conv nets with an interest in understanding how and why they work so well and may also generate additional future work. That being said, there were a number of questionable elements of the work which are illustrated below. Novelty: To my knowledge, the formulation of the RM and DRM are themselves novel and the derivation (supplementary material) of the equivalence is quite interesting. This is probably the strongest part of the paper. Technical Correctness: There is a relatively large assumption, or approximation is Proposation A.1 (a). This is not equivalent, but rather an approximation. I appreciate that the authors explain that this is in a noise-free limit of the GRM, but whether or not this is true in theory makes ALL the difference in practice. Any practical conclusions made by the paper are suspect if the assumptions made by the work are not true in practice... Nit: the nuisance variables are 'g' in the paper but 'h' in the supplementary material. This is quite confusing. Proposition A.1 (b) might be true for a single layer model where it has 3 channels as inputs but it will cease to be true in higher levels of the model. There is a general claim being made between the equivalencies of the DRM and convnets. However, there is a very particular case of convents being compared to, and that is a convnet composed of (only) batch norm layers followed by conv layers, followed by max pooling layers, followed by relus. Oddly this doesn't actually match three of the more popular conv nets (AlexNet, VGG and GoogleNet) which do not use this configuration exactly (AlexNet uses local contrast normalization, VGG uses multiple convs in a row before pooling, GoogleNet uses Inception Modules which are combinations of various branches which are concatenated. How is the image divided into patches? Stride 1 overlapping patches? How are boundary patches used? The order of where batch-norm goes is slightly off. Most models perform batch norm AFTER the operation. Its true that in a deep model, this distinction is not meaningful (e.g. BN just goes between layers) but it certainly matters in the first and last layer. 144: overall pose g^L: this is an interpretation and needs to be validated experimentally. Evaluation: I really wish any other dataset other than MNIST was used. This dataset has been hacked to bits and the margins in which various models operate in terms of gains or losses of performance are within a handful of samples. Consequently, I genuinely am not sure what to make of the results. That being said, their claim that the model achieves better accuracy (in the abstract) is not correct either (See DropConnect for example) and a 1-layer conv net used to compare to is a bit of a straw-man. For a model motivated by the problem of explicitly modeling nuisance variables, it would be much better to use a dataset whose nuisance variables are clearer. What are the nuisance variables for MNIST? Additionally, while the authors do provide a few qualitative results of the generative model (in the supplementary materials), they make a large and ambitious claim about WHAT the model is doing (providing finer and finer details of the image over each layer), but they make no attempt to validate this interpretation. Indeed, even if the DRM and DCN are equivalent in some way, they seem to have traded one problem (why do convents work so well) for another (what is each layer in the DRM really doing). For example, line 167, directly argues about the coarse-to-fine interpretation, but this needs to be validated. Overall, the paper is quite interesting and worth discussion but there seem to be a large number of assumptions made which aren't true and the evaluation could be way better. I'm on the fence regarding acceptance.

Confidence in this Review

2-Confident (read it all; understood it all reasonably well)


Reviewer 3

Summary

The goal of the paper is to provide a theoretical framework for deep learning, especially the modern convolutional neural network. The core of the paper is a factor analysis model whose matrix is factorized into matrices at multiple layers, and the matrix at each layer is chosen from a finite set of multiple alternatives. The model is learned by EM-type algorithm. The method is illustrated by experiment on MNIST.

Qualitative Assessment

The authors seek to develop a theoretical framework to understand deep learning. The proposed multi-layer mixture of factor analysis model and the associated learning and inference algorithm are interesting and potentially useful. The generative model in the paper is a factor analysis model of the form I = LZ + e, where L is the product of matrices at multiple layers (ignoring the bias terms), and each matrix has multiple choices. For EM algorithm, the E-step involves both E(Z) and E(ZZ'), as well as expectation with respect to the multiple choices. The authors suggest to use hard-EM. But it can be problematic to discard the inferential uncertainties in the latent Z and the multiple choices. Even in hard-EM, we need to maximize the log posterior |I-LZ|^2/2sigma^2 + |Z|^2 + constant. Given L, Z should be obtained by penalized least squares that involves sigma^2, which is the marginal variance of e. I do not see this in the algorithm in the main text as well as the supplementary materials. I feel it is not just a simple pseudo-inverse. Even if it is pseudo-inverse, the computation of the pseudo-inverse can be expensive. This is particularly true in the convolutional version, where the multiple choices need to be made at all the relevant locations and all the layers, so that the pseudo-inverse has to be computed on-line in the inference step. The comparison between EM and back-propagation in terms of the number of iterations may be misleading if we do not count the cost of each iteration such as pseudo-inverse computation. To cut to the point, how do the authors use their model to learn from, say, a category of real images in imagenet? Can the model be learned efficiently? Can the model generate realistic images? A minor issue is about the probabilities of multiple choices at multiple layers. Are they estimated? Do they play a role in inference (as they should)?

Confidence in this Review

3-Expert (read the paper in detail, know the area, quite certain of my opinion)


Reviewer 4

Summary

This paper develops a probabilistic framework for convolutional neural networks. Specifically, it develops a DRM-based generative model and shows that operations in CNNs conform to inference in the generative model. The paper then shows that an EM algorithm can replace backpropagation training methods.

Qualitative Assessment

Technical quality: The system model and mathematical development seem correct (although some assumptions are quite restrictive and only mentioned in the supplementary material). My main concern is the experiments section which seems lacking: - The authors perform experiments on the MNIST dataset only -- which is significantly easier than most modern classification datasets. - The experiments only concern the shallow rendering model and do not include the DRM. - In table 1, the first three experiments show that EM training outperforms standard training methods. However, the last experiment shows that related training methods outperform the authors' EM algorithm. The authors claim that the performance of their method is comparable to the other methods. By the same reasoning, standard training methods perform comparably to EM on the first three experiments, as well. The authors' claim of performance increase is therefore questionable. Moreover, the authors attribute the performance drop in the last experiment to the lack of regularization and hyperparameter search, which could have easily been performed. Novelty / Impact: The authors develop a new probabilistic theory that helps give some insights into why CNNs work. They address an important question in deep learning and develop a new training algorithm. This has potential to have a significant impact; however, the experiments do not sufficiently support the authors' claim. Clarity: The paper is well written.

Confidence in this Review

2-Confident (read it all; understood it all reasonably well)


Reviewer 5

Summary

In this paper, the author developed a probabilistic framework, deep rendering model (DRM), to explain the success of the deep convolutional neural nets. Specifically, the max-sum message passing inference procedure in DRM can be recognized as the operations in deep convolution neural nets. Such connection provides an new perspective to understanding the convolution neural nets, both its success and shortcomings, therefore, suggests a route to improve the CNN. Moreover, an EM algorithm is proposed as an alternative for training. Empirical study on MNIST demonstrates the algorithm is promising.

Qualitative Assessment

I like to read this paper which is well-organized and easy to follow. The equivalence between inference in DRM and CNN is justified. The proposed EM algorithm is compared to related work, demonstrating its advantages. There are only several minors which is not clear. 1, several important related work is not suitable cited. The discussion about the relation between inference and CNN have been discovered in many papers, e.g., [1,2,3]. Please specify the difference between these methods to clarify the position of the proposed model and algorithm. 2, I am not clear how Eq.(5) is derived from model defined in Eq. (4). Based on Eq. (4), the observation I follows Gaussian distribution with mean \mu_{c^Lg}. The variance is \sigma^2I, which means the noise are independent. Therefore, the log p(I|\mu, \sigma) propotional to exp(-||I - \mu_{c^Lg}||^2/(2sigma^2)). How the inverse of local nuisance tranformations come out is not clear. 3, In the experiments, it is not clear what are the competitors since the abbreviations of the competitors are not defined. Please add the explanation about the competitors. 4, In appendix, line 428, why the bias in definition of *_{DCN} is \ln\frac{p(a=keep)}{p(a=drop)} rather than b_{cg}? Finally, there are many typos in the manuscript, 1, several references are missing, e.g, line 58 and in appendix table 3. 2, In line 96, figure 3.1A is not found. [1], Zheng, S., Jayasumana, S., Romera-Paredes, B., Vineet, B., Su, Z., Du, D., Huang, C., and Torr, P. Conditional random fields as recurrent neural networks. arXiv preprint arXiv:1502.03240, 2015. [2], Lin, G., Shen, C., Reid, I., and van den Hengel, A. Deeply learning the messages in message passing inference. In NIPS, 2015. [3], Dai, H., Dai, B., and Song, L. Discriminative Embeddings of Latent Variable Models for Structured Data. In ICML, 2016.

Confidence in this Review

2-Confident (read it all; understood it all reasonably well)